# A Preliminary Attempt at the Identification and Financial Estimation of the Negative Health Effects of Urban and Industrial Air Pollution Based on the Agglomeration of Gdańsk

**Piotr O. Czechowski** [1,*], **Piotr Dąbrowiecki** [2], **Aneta Oniszczuk-Jastrząbek** [3], **Michalina Bielawska** [4], **Ernest Czermański** [3], **Tomasz Owczarek** [1], **Patrycja Rogula-Kopiec** [5] **and Artur Badyda** [6]

[1] Faculty of Entrepreneurship and Quality Science, Department of Management and Economics, Department of Quantitative Methods and Environmental Management, Gdynia Maritime University, 81-225 Gdynia, Poland; t.owczarek@wpit.umg.edu.pl

[2] Military Institute of Medicine, Clinic of Infectious Diseases and Allergology, 00-144 Warsaw, Poland; pdabrowiecki@wim.mil.pl

[3] Faculty of Economics, University of Gdańsk, 81-824 Sopot, Poland; ekoao@ug.edu.pl (A.O.-J.); e.czermanski@ug.edu.pl (E.C.)

[4] Agency of Regional Monitoring Atmosphere of Gdańsk Agglomeration, Brzozowa 15 A, 80-243 Gdańsk, Poland; michalina@armaag.gda.pl

[5] Institute of Environmental Engineering, Polish Academy of Sciences, 41-819 Zabrze, Poland; patrycja.rogula-kopiec@ipis.zabrze.pl

[6] Faculty of Building Services, Hydro- and Environmental Engineering, Warsaw University of Technology, Department of Informatics and Environment Quality Research, 00-653 Warsaw, Poland; artur.badyda@pw.edu.pl

[*] Correspondence: p.o.czechowski@wpit.umg.edu.pl; Tel.: +48-502-085-307

**Abstract:** This article marks the first attempt on Polish and European scale to identify the relationship between urban and industrial air pollution and the health conditions of urban populations, while also estimating the financial burden of incidence rates among urban populations for diseases selected in the course of this study as having a causal relation with such incidence. This paper presents the findings of a pilot study based on general regression models, intended to explore air pollutants with a statistically relevant impact on the incidence of selected diseases within the Agglomeration of Gdańsk in the years 2010–2018. In discussing the city's industrial functions, the study takes into consideration the existence within its limits of a large port that services thousands of ships every year, contributing substantially to the volume of emissions (mainly $NO_x$ and PM) to the air. The causes considered include the impact of air pollution, seasonality, land- and sea-based emissions, as well as their mutual interactions. All of the factors and their interactions have a significant impact ($p \leq 0.05$) on the incidence of selected diseases in the long term (9 years). The source data were obtained from the Polish National Health Fund (NFZ), the Agency for Regional Monitoring of Atmosphere in the Agglomeration of Gdańsk (ARMAAG), the Chief Inspectorate of Environmental Protection (GIOŚ), and the Port of Gdańsk Harbourmaster. The study used 60 variables representing the diseases, classified into 19 groups. The resulting findings were used to formulate a methodology for estimating the financial burden of the negative health effects of air pollution for the agglomeration, and will be utilized as a reference point for further research in selected regions of Poland.

**Keywords:** air pollution; health effects of air pollution; statistics; GRM model; COPD; costs of health loss by air emission; asthma; respiratory infections

## 1. Introduction

### 1.1. Aim of the Research

The main object of this paper is to precisely identify the adverse health effects of air pollution reported in urban and industrial agglomerations, both in quantitative terms (incidence) and financial terms (cost burden of medical treatment) in the region's sustainable development policy.

To accomplish this objective, a precise identification has to be made of the diseases related to air pollution and their causes. At this stage, the research project focuses on the air pollution aspect. The problem of air pollution has remained unnoticed for many years, leading to an accumulation of hazards with a direct health impact, such as smog episodes. The cause-and-effect mechanism behind this phenomenon is not adequately studied, with the literature only naming potential causes but providing little indication as to their long-term influence on human health.

### 1.2. Motivation

Polish cities with different degrees of air pollution problems, such as Warsaw [WAW], Tricity (Gdańsk, Sopot and Gdynia) [TRJ], Cracow [KRA], Zabrze [ZAB] and Nowy Sacz [NSA], have been selected as case studies for the identification of statistically relevant factors behind particularly health-threatening diseases. A relatively long nine-year period (2010–2018) has been considered so as to identify the direct cause-and-effect relationships, as well as determine the long-term effects of exposure to a polluted environment, while also taking into account the interactions between pollutant concentrations, weather conditions and time of exposure. The results for Tricity will set the stage for studies on other urban areas. Besides the reasons discussed above, Tricity has been selected as a pilot case study because of its seaside location and the influence of other factors associated with sea port emissions, in particular, those attributable to ships (engine exhaust emissions from arriving/departing ships and exhaust from power generators in ships moored at berth).

A precise identification of the factors requires the use of statistical tools. Of these, the most fundamental one is the reference grid of the automatic air monitoring system. Of all the cities concerned, the most extensive grid is found in Tricity. The grid should ensure that measurements are reliable so as to enable the data derived from them to be utilized for exploring models of health impact with key implications for city residents. Statistical methodology is of prime importance for this purpose [1].

The use of quantitative methods, including stochastic and exploratory techniques, in environmental studies does not seem to be sufficient for practical purposes. There is no comprehensive dedicated analytical system to address this issue, or research regarding this subject. The methodological emphasis at the initial stage of work was placed on data quality assessment through the authors' own data quality method [2]—using harmonic models and robust estimators in addition to the classical tests of outlier values with their iterative expansions. The results obtained demonstrate both the complementarity of the proposed solution in relation to classical methods as well as allowing a significant extension of the range of applications. The practical usefulness is also highly significant due to the high effectiveness and numerical efficiency as well as the simplicity of this new tool.

### 1.3. Global Background and Local Air Quality Problems

According to UNFPA (United Nations Population Fund), the world's population reached 7.715 billion in 2019. On a global scale, urban residents make up over 50% of the overall worldwide population. The same figure stands at approximately 75% for Europe alone. Moreover, estimates show that, by 2030, the world population (which will by then reach 8.5 billion) will include 5 billion city dwellers (over 60%) [3]. This means that the maintenance of air quality, especially in large urban areas, will be an increasingly serious challenge for institutions and governing bodies managing the quality of the environment. In various locations in the world, as indicated by the World Health Organization's data (WHO Global Ambient Air Quality Database), air quality deviates significantly not only from the rather restrictive WHO guidelines, but also from the usually more liberal local legal

regulations. As a result, 92% of the world's population lives in conditions where the WHO standards are exceeded [4]. This in turn causes ambient air pollution to account for an estimated 4.2 million deaths per year due to stroke, heart disease, lung cancer and chronic respiratory diseases. Although the most unfavorable situation applies to some Asian and African countries and the Middle East, Poland is one of the most polluted countries in the European Union.

As a consequence of the relatively high emissions of air pollutants in Poland, limit values of particulate matter ($PM_{10}$ and $PM_{2.5}$) concentrations (according to 2004/107/EC Directive) [5] as well as benzo(a)pyrene (BaP) target values (according to 2008/50/WE Directive) [6] are regularly exceeded. In some areas (a relatively small number (4–6) of locations), calendar year limit values for nitrogen dioxide ($NO_2$) as well as target values for ozone ($O_3$) and arsenic (As) are also not complied with. The principal sources of air pollutant emissions to ambient air include the municipal and household sector as well as road transport. According to the European Environmental Agency's (EEA) Air Pollutant Emissions Data Viewer (most recent data from 2017), the municipal and household sector in Poland in 2017 was mostly responsible for the emission of $PM_{10}$, $PM_{2.5}$ and carbon monoxide (CO). The annual emission of PM10 was 125,082 tons (Mg) (which is 50.8% of the total national emission), of $PM_{2.5}$ was 78,937 tons (53.6% of the total emission), and of CO was 1,598,900 tons (62.9% of the total emission). The highest share of this sector in the total emission balance concerns the benzo(a)pyrene–commercial and household sector, due to the fact that solid fuel (mainly coal and wood) incineration is responsible for 83.6% of the total BaP national emission (i.e., 34 tons). Meanwhile, road transportation is primarily responsible for nitrogen oxides (NOx) and CO emission. The annual national emission of NOx in 2017 accounted for 297,356 tons (which is 37.0% of the total national emission), and of CO was 588,444 tons (23.1% of the total emission). These two sectors of the Polish economy overwhelmingly shape the air quality, although the impact on the so-called background concentration also involves the sectors of energy production and distribution as well as industrial processes and product use.

A quantitative (model) identification of diseases arising from long-term exposure (more than 9 years) to air pollution has never been made in Poland. Such identification will allow the estimation of the actual financial burden of air pollution to society.

This paper is an interdisciplinary project addressing three main areas of concern: health, environment and economy. The three aspects are interconnected from a statistical perspective and hard-wired into information systems currently under construction. The wide thematic scope of this project will allow us to address only certain Gdańsk-specific questions with key implications for the achievement of the research objectives set out in this paper.

### 1.4. Social Background of the Issue and Literature Review

This section focuses on air pollution as one of the most dangerous environmental impacts on the development and functioning of the respiratory system. The key respiratory diseases include: asthma, chronic obstructive pulmonary disease (COPD), and respiratory infections.

Research shows that both short- and long-term exposure to common air pollutants at elevated concentrations is associated with heightened incidence and mortality rates for respiratory diseases [7,8]. One of the key pollutants with adverse effects on the respiratory tract is suspended particulate matter. Depending on particle size, suspended matter may penetrate various parts of the respiratory tract. Particulate matter deposits in the upper parts of the respiratory tract may aggravate the symptoms of asthma and COPD [9]. Water-soluble gaseous pollutants (e.g., $SO_2$) are absorbed mainly in the upper parts of the respiratory tract, promoting damage to upper airways and primary bronchi. Gases with lower water solubility (e.g., $NO_2$ and $O_3$) mainly affect the lower respiratory tract [10].

Bronchial asthma is estimated to affect approximately 235 million people globally, causing 345 thousand deaths every year [11]. The sharp increase in the worldwide incidence of asthma, especially in industrialized countries, has made it the most frequent chronic children's disease [12].

Chronic obstructive pulmonary disease (COPD) is characterized by partially irreversible restriction of airflow through the respiratory tract, triggering an inflammatory response to various harmful

substances [13]. COPD is a major problem in developing and developed countries alike. Estimates suggest that, in 2020, the condition will have become the world's third leading cause of death and the fifth cause of motor impairment or even disability, generating high social and economic costs [14]. Poland has a high number of COPD and asthma sufferers (estimated at approx. 6 million in total), 80% of whom have not been adequately diagnosed and therefore not offered proper medical attention [15,16]. Smoking tobacco remains the largest risk factor for COPD, accounting for 80% of instances of this illness [14]. This is followed by exposure to occupational hazards and to polluted air [14,17]. However, COPD also affects non-smokers. Research conducted in the USA under the NHANES III project found that 19.2% of diagnosed cases of COPD among 10 thousand adults aged 30–75 were attributable to exposure to polluted air in the workplace. In the sample of non-smokers, exposure to occupational hazards accounted for 31.1% of cases of COPD [18]. The increase in COPD incidence worldwide cannot be satisfactorily accounted for solely by smoking tobacco without regard to any other factors [17].

Exposure to air pollutants substantially increases the incidence of respiratory infections, including pneumonia, especially in children [19] and the elderly [20]. It is noteworthy that pneumonia is among the leading causes of death in developed countries. As for the elderly, some studies argue that a link exists between short-term exposure to air pollutants and incidence of pneumonia [21,22]. Long-term exposure to air pollutants has also been shown to be a risk factor for respiratory infections. A survey conducted in Hamilton (Canada) on subjects aged over 65 revealed a correlation between heightened exposure to nitrogen dioxide and/or PM2.5 and an increased number of hospitalizations for pneumonia [22,23].

*1.5. Economic Background of the Issue and Literature Overview*

The economic effects of health loss due to diseases related to air pollution may be studied from a number of perspectives: financial (lost earnings), social (lost GDP), social insurance (pay-out on health insurance and disability pensions), and taxpayer (National Health Fund, Ministry of Health). The analysis considers direct (medical and non-medical) costs, indirect costs, and social costs representing the total burden to the patient and to the economy as a whole. Direct costs are financial burdens to society in the form of money transfers from the healthcare system to entities providing medical services. This cost group represents the main cost component of illness, as it takes the form of cash transfers flowing from the National Health Fund to hospitals or from patients to hospitals. These are not the only costs of illness, however. There are also indirect costs, which make up more than half of total medical costs [24], broadly defined as production losses [25]. Additionally, the indirect costs component also includes the costs of out-of-system medical care, costs of free-of-charge labor, compensation mechanisms, and the group dependency effect [26]. It is noteworthy that indirect costs of medical care are usually related to disease itself, while direct costs are usually associated with the treatment process or preventive measures. Therefore, by footing direct costs, it is possible to reduce indirect costs.

Due to a lack of financial data on procedures funded by the National Health Fund, this article will focus on the first group of costs, i.e., direct costs of medical treatment.

## 2. Materials and Methods

Primary data drawn from the following sources have been used to construct statistical models:

Health data—National Health Fund database covering the period from 1 January 2010 to 31 December 2018 on all health services rendered in Poland by region (14,387,846 services).

Air pollution data–hourly data for the Tricity area collected from five measurement stations located within the Agglomeration of Gdańsk (AM2, AM3, AM5, AM6 and AM8) for the period from 2010 to 2018 (Figure 1).

The stations are located close to the Bay of Gdańsk. This study includes gaseous pollutants (sulfur dioxide, nitrogen dioxide, nitrogen oxides, ozone, carbon oxide and carbon dioxide), particulate pollutants ($PM_{10}$ and $PM_{2.5}$) as well as weather parameters (temperature, relative air humidity, wind

force and direction, and rainfall). Additional data on benzo(a)pyrene from the Chief Inspectorate of Environmental Protection's manually operated station in Gdańsk, covering the period from 1 January 2010 to 21 December 2018, have been copied from the air quality website at http://powietrze.gios.gov.pl/pjp/archives.

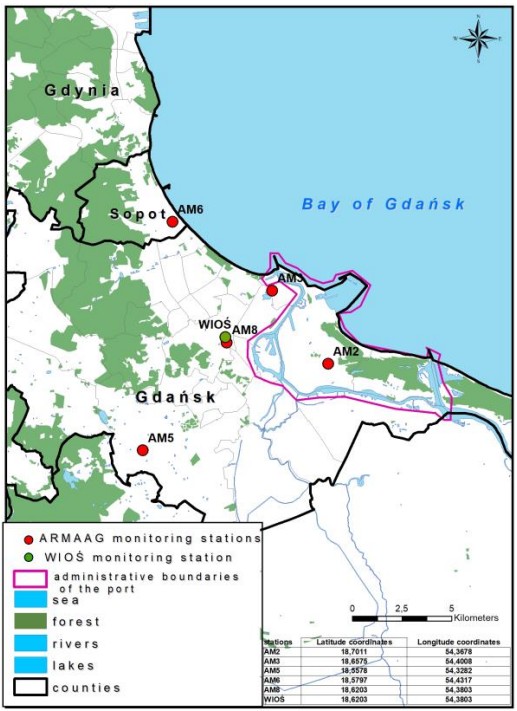

**Figure 1.** Location of measurement stations within the Agency of Regional Monitoring of Atmosphere in the Agglomeration of Gdańsk (ARMAAG) network utilized in the study.

Ship traffic data come from the port of Gdańsk (54°25′ N, 18°39′ E) and cover the period from 1 January 2010 to 6 November 2017.

All of the data have been entered into a single database after being counted and added up (health data, ship traffic data) or averaged (Agency for Regional Monitoring of Air Pollution, Chief Inspectorate of Environmental Protection). The article utilizes a number of statistical methods and models (analysis of variance, ANOVA; analysis of co-variance, ANCOVA; Cluster Analysis, CA; Principal Component Analysis models, PCA; and others), of which GRMs (Generalized Regression Models) are the most important.

GRMs are an extension of the GLM (Generalized Linear Model) family. A general linear model may be treated as an extension of multiple linear regression for a single dependent variable. The multiple regression model underlies general linear regression. The general purpose of multiple regression (a term first used by Pearson in 1908) is to give a qualitative overview of relationships between multiple independent (controlled, explanatory) variables and dependent (criterion, explained) variables.

The basic multiple regression model in its general form is as follows:

$$Y = b_0 + b_1 X_1 + b_2 X_2 + \cdots + b_k X_k \tag{1}$$

where:

$Y$—explained variable;
$k$—number of predictors (controlled variables).

Contrary to the multiple regression model, which is more suitable for analyzing continuous predictors (strong measurement scales, such as weather measurements or pollutant concentration

measurements), the general linear model can be more readily applied to any instance of analysis of variance (ANOVA) featuring qualitative (categorized) predictors, to any instance of analysis of co-variance (ANCOVA) featuring qualitative (categorized) predictors, such as heating periods or rainfall, as well as any model of regressive analysis featuring continuous predictors.

In the case of qualitative predictors, the outcomes may be coded in experiment matrix X, using a re-parameterized model or a sigma-limiting model.

Contrary to other models, the Generalized Linear Regression Model is not a model in a strict sense, but a modeling pathway comprising a variety of model classes and estimation methods:

- Simple regression
- Multiple regression
- Factor regression
- Polynomial regression
- Response surface regression
- Response surface regression for mixtures
- Single-factor ANOVA
- Main effects ANOVA
- Factorial ANOVA
- Analysis of co-variance (ANCOVA)
- Identical slopes model

GRM consolidates all these models and allows for identification of a cause-and-effect relationship regardless of the measurement scale of independent variables.

The first step in model quality assessment is to verify how well empirical data fit into a model, that is to say, to test the goodness of fit, with the available error measures applied. The most commonly used metric for good fit assessment is the determination co-efficient:

$$R^2 \; = \; 1 - \frac{n-1}{n-k-1}\left(1 - \frac{\sum_{t=1}^{n}(\hat{x}_t - \overline{x})^2}{\sum_{t=1}^{n}(x_t - \overline{x})^2}\right) \tag{2}$$

where:

$x_t$—values of variable X at time or period t,
$\hat{x}_t$—theoretical value of variable X at time or period t,
$\overline{x}$—mean value of variable X in a time series on n observations,
$n$—number of observations,
$k$—number of explanatory variables.

This metric shows the goodness of the model's fit with the empirical data. Its primary advantage is normalization. In fact, the metric is simply an adjusted coefficient of determination based on the "penal factor," favoring multivariate models with fewer independent variables. In this study, this metric also plays an explanatory role in that it specifies what portion of the information pool on disease incidence variability could be accounted for by the model and therefore also to what extent the identified predictors account for total disease incidence variability.

The evaluation of model errors, expressed as root mean square of the error, holds a central place in model quality assessment:

$$\mathrm{RMSE} \; = \; \sqrt{\mathrm{MSE}} \; = \; \sqrt{\frac{1}{n}\sum_{t=1}^{n} e_t^2} \tag{3}$$

where:

$e_t \; = \; x_t \, - \, \hat{x}_t$—model remainders.

This metric shows how far the actual values deviate from theoretical values determined in the model. Useful, though not always accurate, information can also be extracted from the random-error-based variability coefficient reflecting the mean level of the phenomenon:

$$V(S_e) \;=\; \frac{\text{RMSE}}{\overline{x}} \cdot 100 \tag{4}$$

The formula gives an idea of the root mean square error concentration within the medium level. AICC (Akaike Information Criterion with correct for small size sample) criteria are generalized FPE (Akaike's Final Prediction Error Criterion) criteria proposed by Akaike.

$$\text{AICC}(\beta) : -2lnL_x\left(\beta, \frac{S_x(\beta)}{n}\right) + \frac{2(p+q+1)n}{(n-p-q-2)} \tag{5}$$

The AICC criteria differ from AIC (Akaike Information Criterion) in weighting adjustment. AIC and AICC statistics are based on the quotient of the maximum likelihood function. Criterion design is done by comparing the estimated model with the full (ideal) model. The model with the lowest criterion value is thought to be the best.

Eventually, the formula proposed by Makridakis as an extension of the previous variants was adopted as the key selection criterion:

$$AICC \;=\; n(1 + \log(2\pi)) + n \cdot ln\hat{\sigma}_k^2 + 2k \tag{6}$$

The calculation methodology adopted in the economic section comprises aggregated cost categories, such as medical care costs, hospitalization costs, diagnostic costs, medication costs and costs of specialist consultancy services provided to in-patients in all hospitals in Gdańsk and Gdynia within the period in question. This breakdown into five cost categories is consistent with the standards for calculation of medical treatment costs adopted by Polish medical service providers as the basis for charging fees for contracted treatments. The value data have been drawn from a register of services subsidized by the National Health Fund and provided by the Clinic of Infectious Diseases and Allergology at the Military Institute of Medicine in Warsaw in 2018 for diseases selected in the course of this study as being correlated with air pollution in urban agglomerations. The classification includes a calculation of total medical costs of specific diseases, expressed as the combined products of lump-sum financial costs disclosed by the service provider. The data have been additionally refined by the inclusion of refund amounts paid monthly by the National Health Fund to medical service providers for each completed procedure relating to diseases covered by the register.

## 3. Results

For all the relevant diseases represented by the variables in the first column (Table 1), the percentage of valid observations (column "% Valid Obs") and basic distribution characteristics (asymmetry and kurtosis) have been calculated. The column "standard Normality tests" contains the results of a Kolmogorov–Smirnov [K–S] test, a K–S with Lilliefors correction and of a Shapiro–Wilk Francia test with Royston correction. The columns "Distribution 1st Similar" and "Distribution 2nd Similar" contain estimation results by the Maximum Likelihood estimation of the two most similar distributions based on the Minimum Likelihood Criterion. The tests have show a deviation from the normal distribution of variables without their levels being considered. For all variables, the Normal distribution (with possible Box–Cox logarithmic transformations) is the best empirical approximation of the investigated variables; therefore, it has been assumed that the distributions of empirical variables are largely similar to the normal distribution.

The next step was to identify, for each disease, factor models in relevant correlation with incidence rates for a specific disease (Table 2) without singling out emissions of maritime origin and factor models considering only sea winds (emissions of maritime origin) (Table 3). A comparison of outcomes derived from the two models leads to clear conclusions regarding emissions of maritime or port origin with health impact on urban populations.

**Table 1.** Selected numerical data on the incidence of diseases within the Agglomeration of Gdańsk [TRJ] in the years 2010–2018 (an abbreviations list of the names of Variables can be found in Appendix A); abbreviations: LogN, LogNormal; ExtremeV, Extreme Value.

| Variable | Valid N | % Valid Obs. | Skewness | Kurtosis | Standard Normality Tests | Distribution 1st Similar; Likelihood | Distribution 2nd Similar; Likelihood |
|---|---|---|---|---|---|---|---|
| TRJ_R00 | 2123 | 64.6 | 1.6 | 2.8 | K–S d = 0.35507. $p < 0.01$ <br> Lilliefors $p < 0.01$ <br> Shapiro–Wilk W = 0.69944. $p = 0.00$ | LogN10; −3,640,962 | LogN; −3,640,962 |
| TRJ_R05 | 1624 | 49.4 | 1.8 | 3 | K–S d = 0.43852. $p < 0.01$ <br> Lilliefors $p < 0.01$ <br> Shapiro–Wilk W = 0.59506. $p = 0.00$ | ExtremeV; −4,360,485 | LogN; −2,462,986 |
| TRJ_R06 | 2475 | 75.3 | 1.3 | 1.9 | K–S d = 0.30496. $p < 0.01$ <br> Lilliefors $p < 0.01$ <br> Shapiro–Wilk W = 0.75174. $p = 0.00$ | Weibull; −18,893,110 | Normal; −18,850,110 |
| TRJ_R07 | 3287 | 100 | 0.6 | 0.5 | K–S d = 0.12878. $p < 0.01$ <br> Lilliefors $p < 0.01$ <br> Shapiro–Wilk W = 0.96351. $p = 0.00$ | ExtremeV; -30,175,610 | Normal; −30,030,220 |
| TRJ_sum_J00_J06 | 3286 | 100 | 0.6 | 0.7 | K–S d = 0.05410. $p < 0.01$ <br> Lilliefors $p < 0.01$ <br> Shapiro–Wilk W = 0.97683. $p = 0.00$ | LogN; −30,985,510 | LogN10; −30,985,510 |
| TRJ_sum_J12_18 | 3283 | 99.9 | 0.6 | 0.4 | K–S d = 0.08876. $p < 0.01$ <br> Lilliefors $p < 0.01$ <br> Shapiro–Wilk W = 0.96812. $p = 0.00$ | LogN10; −18,148,780 | LogN; −18,148,780 |
| TRJ_sum_j20_j22 | 3151 | 95.9 | 1 | 1.6 | K–S d = 0.14026. $p < 0.01$ <br> Lilliefors $p < 0.01$ <br> Shapiro–Wilk W = 0.91796. $p = 0.00$ | LogN; −10,771,570 | LogN10; −10,771,570 |
| TRJ_sum_J31_J34 | 2943 | 89.5 | 0.6 | 0.2 | K–S d = 0.09412. $p < 0.01$ <br> Lilliefors $p < 0.01$ <br> Shapiro–Wilk W = 0.95947. $p = 0.00$ | LogN; −18,465,880 | LogN10; −18,465,880 |
| TRJ_sum_J37_J39 | 2528 | 76.9 | 1.1 | 1.3 | K–S d = 0.18902. $p < 0.01$ <br> Lilliefors $p < 0.01$ <br> Shapiro–Wilk W = 0.86093. $p = 0.00$ | LogN10; -6,469,712 | LogN; −6,469,712 |
| TRJ_sum_J40_J42 | 1062 | 32.3 | 2.1 | 4.8 | K–S d = 0.36616. $p < 0.01$ <br> Lilliefors $p < 0.01$ <br> Shapiro–Wilk W = 0.66306. $p = 0.00$ | ExtremeV; −1,781,786 | Normal; −1,747,864 |
| TRJ_sum_J43_J44 | 3141 | 95.6 | 1 | 1.4 | K–S d = 0.21734. $p < 0.01$ <br> Lilliefors $p < 0.01$ <br> Shapiro–Wilk W = 0.88517. $p = 0.00$ | LogN; −6,702,670 | LogN10; −6,702,670 |
| TRJ_sum_J45_J46 | 3122 | 95 | 0.7 | 0 | K–S d = 0.13202. $p < 0.01$ <br> Lilliefors $p < 0.01$ <br> Shapiro–Wilk W = 0.94005. $p = 0.00$ | LogN10; −14,354,210 | LogN; −14,354,210 |

**Table 1.** *Cont.*

| Variable | Valid N | % Valid Obs. | Skewness | Kurtosis | Standard Normality Tests | Distribution 1st Similar; Likelihood | Distribution 2nd Similar; Likelihood |
|---|---|---|---|---|---|---|---|
| TRJ_sum_I21_I23 | 3256 | 99.1 | 0.7 | 0.6 | K–S d = 0.18745. $p < 0.01$<br>Lilliefors $p < 0.01$<br>Shapiro–Wilk W = 0.92258. $p = 0.00$ | LogN;<br>−9,236,908 | LogN10;<br>−9,236,908 |
| TRJ_sum_I63_I64 | 3267 | 99.4 | 0.7 | 0.4 | K–S d = 0.13726. $p < 0.01$<br>Lilliefors $p < 0.01$<br>Shapiro–Wilk W = 0.94685. $p = 0.00$ | LogN10;<br>−9,481,207 | LogN;<br>−9,481,207 |
| TRJ_sum_I65_I66 | 2130 | 64.8 | 1.2 | 1.1 | K–S d = 0.23949. $p < 0.01$<br>Lilliefors $p < 0.01$<br>Shapiro–Wilk W = 0.84043. $p = 0.00$ | LogN10;<br>−5,104,719 | LogN;<br>−5,104,719 |
| TRJ_J11 | 213 | 6.5 | 2.7 | 10 | K–S d = 0.46368. $p < 0.01$<br>Lilliefors $p < 0.01$<br>Shapiro–Wilk W = 0.51370. $p = 0.00$ | ExtremeV;<br>−333,195 | LogN;<br>−332,489 |
| TRJ_J30 | 303 | 9.2 | 3.1 | 9 | K–S d = 0.53048. $p < 0.01$<br>Lilliefors $p < 0.01$<br>Shapiro–Wilk W = 0.33165. $p = 0.00$ | ExtremeV;<br>−400,824 | Normal;<br>−394,480 |
| TRJ_J47 | 751 | 22.8 | 2.7 | 6.4 | K–S d = 0.52151. $p < 0.01$<br>Lilliefors $p < 0.01$<br>Shapiro–Wilk W = 0.38298. $p = 0.00$ | LogN;<br>−1,050,949 | LogN10;<br>−1,050,949 |
| TRJ_I20 | 3110 | 94.6 | 0.6 | 0.2 | K–S d = 0.15689. $p < 0.01$<br>Lilliefors $p < 0.01$<br>Shapiro–Wilk W = 0.93295. $p = 0.00$ | LogN10;<br>−8,957,948 | LogN;<br>−8,957,948 |

**Table 2.** List of significant ($p \leq 0.05$) factors with relevant impact on disease incidence within the Agglomeration of Gdańsk [TRJ], without considering the influence of sea wind (list of abbreviations in Appendix A, detailed model results in Table 5); model F test statistics and *p*-level.

| Disease | %Var | Interactions | | | | | | F | p |
|---|---|---|---|---|---|---|---|---|---|
| TRJ_R00 | 7.7% | YYYY * ShipNo | MM * RAIN | $NO_2$ | | | | 5.59 | 0.00 |
| TRJ_R05 | 31.2% | RAIN * $O_3$ | MM * $PM_{2.5}$ | DD * YYYY | MM * TEMP | | | 1.49 | 0.00 |
| TRJ_R06 | 30.1% | $SO_2$ * $O_3$ | $O_3$ * ShipNo | YYYY * $O_3$ | DD * MM | YYYY * TEMP | YYYY * WV | 1.41 | 0.00 |
| TRJ_R07 | 11.1% | $SO_2$ * $O_3$ | YYYY * CO | ShipNo | NO2 * WV | CO * PRES | $NO_X$ * HUMID | 12.52 | 0.00 |
| | | NO * ShipNo | YYYY * WV | | | | | | |
| TRJ_sum_J00_J06 | 56.0% | MM | DD * MM | MM * YYYY | YYYY * SO2 | YYYY * $O_3$ | $NO_2$ * $O_3$ | 4.52 | 0.00 |
| | | $O_3$ * CO | MM * $PM_{2.5}$ | $CO_2$ * TEMP | TEMP * HUMID | WV * ShipNo | $NO_2$ * BaP | | |
| | | CO * BaP | | | | | | | |
| TRJ_J11 | 8.2% | $SO_2$ * HUMID | NO2 * BaP | | | | | 4.96 | 0.01 |
| TRJ_sum_J12_18 | 46.9% | MM | MM * YYYY | NO2 * WV | $PM_{2.5}$ * WV | YYYY*PRES | $PM_{2.5}$ * BaP | 3.28 | 0.00 |
| | | DD * MM | CO | WV*BaP | TEMP * ShipNo | $SO_2$*ShipNo | $NO_2$ * HUMID | | |
| | | $O_3$ * $PM_{2.5}$ | | | | | | | |
| TRJ_sum_j20_j22 | 43.9% | CO * $PM_{2.5}$ | YYYY*NO | MM | MM * YYYY | DD * YYYY | CO * BaP | 3.70 | 0.00 |
| | | DD * BaP | RAIN * $PM_{2.5}$ | RAIN *$O_3$ | | | | | |
| TRJ_J30 | 18.7% | YYYY * WV | RAIN *TEMP | WV*TEMP | | | | 4.39 | 0.00 |
| TRJ_sum_J31_J34 | 10.9% | WV * ShipNo | NO*O3 | MM*WV | YYYY* $SO_2$ | $O_3$*CO | $NO_2$ * $O_3$ | 6.86 | 0.00 |
| | | $PM_{2.5}$* ShipNo | $O_3$* TEMP | RAIN *$PM_{10}$ | YYYY * NO | | | | |
| TRJ_sum_J37_J39 | 12.1% | CO2 | DD * NO * $CO_2$ | YYYY * $PM_{10}$ | NO * WV | YYYY * HUMID | | 4.59 | 0.00 |

**Table 2.** *Cont.*

| Disease | %Var | Interactions | | | | | | F | p |
|---|---|---|---|---|---|---|---|---|---|
| TRJ_sum_J40_J42 | 32.9% | YYYY | WV * BaP | $NO_2$ * TEMP | | | | 37.10 | 0.00 |
| TRJ_sum_J43_J44 | 13.1% | MM | $NO_2$ * ShipNo | YYYY * PM10 | $O_3$ * BaP | | | 2.66 | 0.00 |
| TRJ_sum_J45_J46 | 12.2% | $CO_2$ * ShipNo | $NO_2$ * WV | CO * WV | RAIN * MM | YYYY * $CO^2$ | $O_3$ * PM10 | 6.08 | 0.00 |
| | | $NO_2$ | WV | $NO_2$ * ShipNo | YYYY * ShipNo | MM * $O_3$ | | | |
| TRJ_J47 | 6.1% | YYYY * WV | | | | | | 4.14 | 0.00 |
| TRJ_I20 | 7.8% | ShipNo | YYYY * ShipNo | $NO_X$ * WV | CO * CO2 | $NO_2$ * HUMID | $SO_2$* WV | 11.94 | 0.00 |
| | | YYYY * CO | | | | | | | |
| TRJ_sum_I21_I23 | 15.9% | YYYY * HUMID | MM * YYYY | RAIN * MM | NO2 * ShipNo | TEMP * BaP | $PM_{2.5}$ * ShipNo | 3.65 | 0.00 |
| | | NOX * WV | | | | | | | |
| TRJ_sum_I63_I64 | 3.9% | CO * ShipNo | YYYY * HUMID | NO * TEMP | $PM_{2.5}$ * TEMP | $NO_2$ * $O_3$ | TEMP * BaP | 6.70 | 0.00 |
| TRJ_sum_I65_I66 | 2.3% | TEMP * HUMID | CO * HUMID | RAIN * BaP | TEMP * ShipNo | | | 8.09 | 0.00 |

**Table 3.** List of factors with relevant impact on the incidence of diseases within the Agglomeration of Gdańsk [TRJ], considering the influence of sea wind (list of abbreviations in Appendix A, detailed model results in Table 5); model F test statistics and *p*-level.

| Variables | %Var | Interactions | | | | | | F | p |
|---|---|---|---|---|---|---|---|---|---|
| TRJ_R00 | 31.8% | YYYY * $NO_X$ | $DD * O_3$ | DD * PRES | $O_3 * CO_2$ | | | 2.12 | 0.00 |
| TRJ_R05 | N/O | | | | | | | | |
| TRJ_R06 | 8.2% | YYYY * $O_3$ | CO * TEMP | RAIN * NO | | | | 4.88 | 0.00 |
| TRJ_R07 | 7.1% | YYYY * $NO_2$ | WV * HUMID | YYYY * $NO_X$ | | | | 3.29 | 0.00 |
| TRJ_sum_J00_J06 | 56.7% | YYYY * $PM_{10}$  RAIN * BaP | RAIN * $PM_{10}$  MM * $PM_{10}$ | MM * PRES | $NO_2$ * ShipNo | MM * YYYY | $NO_2$ * NOX | 6.38 | 0.00 |
| TRJ_J11 | N/O | | | | | | | | |
| TRJ_sum_J12_18 | 49.1% | MM * $CO_2$  DD * WV | WV * TEMP  DD* $SO_2$ | YYYY * BaP | MM * BaP | DD * RAIN | $O_3$ * BaP | 3.61 | 0.00 |
| TRJ_sum_j20_j22 | 43.2% | MM  NO * $O_3$ | DD * BaP | YYYY * NOX | YYYY * BaP | ShipNo * BaP | MM * HUMID | 5.50 | 0.00 |
| TRJ_J30 | N/O | | | | | | | | |
| TRJ_sum_J31_J34 | 9.7% | $O_3$ * HUMID | MM * NO | WV * ShipNo | $SO_2$ * BaP | $CO_2$ | NO * $NO_2$ | 3.38 | 0.00 |
| TRJ_sum_J37_J39 | 24.1% | RAIN * BaP | YYYY * PRES | HUMID * ShipNo | DD * WV | | | 2.97 | 0.00 |
| TRJ_sum_J40_J42 | 79.1% | $PM_{2.5}$ * HUMID  RAIN *WV | $PM_{10}$ * $PM_{2.5}$  MM * YYYY | RAIN *PRES*ShipNo | NOX * HUMID | WV*HUMID | RAIN *TEMP | 4.93 | 0.00 |
| TRJ_sum_J43_J44 | 9.6% | TEMP | NOX*WV | RAIN *TEMP | | | | 19.17 | 0.00 |
| TRJ_sum_J45_J46 | 5.9% | WV* ShipNo | $NO_2$*PRES | $PM_{10}$*$PM_{2.5}$ | PRES | RAIN *$PM_{2.5}$ | | 6.83 | 0.00 |
| TRJ_J47 | 57.1% | YYYY*BaP | DD*BaP | YYYY*NO | | | | 2.89 | 0.00 |
| TRJ_I20 | 2.0% | $NO_X$* WV | RAIN *TEMP | | | | | 5.62 | 0.00 |

**Table 3.** *Cont.*

| Variables | %Var | Interactions | | F | $p$ |
|---|---|---|---|---|---|
| TRJ_sum_I21_I23 | 17.6% | $NO_2$ * BaP | MM * YYYY | 1.61 | 0.00 |
| TRJ_sum_I63_I64 | N/O | | | | |
| TRJ_sum_I65_I66 | 6.4% | $O_3$ * WV | $CO_2$ * ShipNo | 12.59 | 0.00 |

"N/O": A stable model could not be identified. Source: authors' own work.

**Table 4.** A list of significant ($p \leq 0.05$) factors with relevant impact on disease incidence within the Agglomeration of Gdańsk [TRJ], without considering the influence of sea wind (list of abbreviations in Appendix A); df, SS (Sum of squares), MS (Mean squares), F statistics and *p*-level of factors and interactions.

| Dependent Variable | Effect | Degr. of Freedom | SS | MS | F | *p* |
|---|---|---|---|---|---|---|
| TRJ_R00 | Intercept | 1 | 7,646,826 | 7,646,826 | 832.70 | 0.00 |
| | YYYY * ShipNo | 7 | 665,979 | 95,140 | 10.36 | 0.00 |
| | MM * TRJ.RAIN Y_N | 11 | 293,390 | 26,672 | 2.90 | 0.00 |
| | TRJ.NO$_2$ | 1 | 46,917 | 46,917 | 5.11 | 0.02 |
| | Error | 1276 | 11,717,736 | 9183 | | |
| | Total | 1295 | 12.693,180 | | | |
| | Total | 1295 | 12,693,180 | | | |
| TRJ_R05 | Intercept | 1 | 84,590,65 | 8,459,065 | 1605.07 | 0.00 |
| | TRJ.RAIN Y_N * TRJ.O$_3$ | 1 | 34,080 | 34,080 | 6.47 | 0.01 |
| | MM * TRJ.PM$_{2.5}$ | 11 | 135,951 | 12,359 | 2.35 | 0.01 |
| | YYYY * DD | 210 | 1,380,290 | 6573 | 1.25 | 0.02 |
| | MM * TRJ.TEMP | 11 | 224,528 | 20,412 | 3.87 | 0.00 |
| | Error | 765 | 4,031,716 | 5270 | | |
| | Total | 998 | 5,857,466 | | | |
| TRJ_R06 | Intercept | 1 | 6,811,330 | 6,811,330 | 629.80 | 0.00 |
| | TRJ.SO$_2$ * TRJ.O$_3$ | 1 | 130,206 | 130,206 | 12.04 | 0.00 |
| | TRJ.O$_3$ * ShipNo | 1 | 66,003 | 66,003 | 6.10 | 0.01 |
| | YYYY * TRJ.O$_3$ | 7 | 259,244 | 37,035 | 3.42 | 0.00 |
| | DD * MM | 330 | 4,285,668 | 12,987 | 1.20 | 0.02 |
| | YYYY * TRJ.TEMP | 7 | 211,663 | 30,238 | 2.80 | 0.01 |
| | YYYY * TRJ.WV | 7 | 192,013 | 27,430 | 2.54 | 0.01 |
| | Error | 1162 | 12,567,202 | 10,815 | | |
| | Total | 1515 | 17,969,271 | | | |
| TRJ_R07 | Intercept | 1 | 36,224,059 | 36,224,059 | 650.03 | 0.00 |
| | TRJ.SO$_2$ * TRJ.O$_3$ | 1 | 1,124,489 | 1,124,489 | 20.18 | 0.00 |
| | YYYY * TRJ.CO | 7 | 1,944,100 | 277,729 | 4.98 | 0.00 |
| | ShipNo | 1 | 682,159 | 682,159 | 12.24 | 0.00 |
| | TRJ.NO$_2$ * TRJ.WV | 1 | 1,170,553 | 1,170,553 | 21.01 | 0.00 |
| | TRJ.CO * TRJ.PRES | 1 | 1,028,956 | 1,028,956 | 18.46 | 0.00 |
| | TRJ.NO$_X$ * TRJ.HUMID | 1 | 839,760 | 839,760 | 15.07 | 0.00 |
| | TRJ.NO * ShipNo | 1 | 219,653 | 219,653 | 3.94 | 0.05 |
| | YYYY * TRJ.WV | 7 | 810,664 | 115,809 | 2.08 | 0.04 |
| | Error | 2011 | 112,066,208 | 55,727 | | |
| | Total | 2031 | 126,018,502 | | | |
| TRJ_sum_J00_J06 | Intercept | 0 | | | | |
| | MM | 7 | 13,252,869 | 1,893,267 | 17.20 | 0.00 |
| | DD * MM | 326 | 46,207,886 | 141,742 | 1.29 | 0.00 |
| | MM * YYYY | 77 | 31,335,749 | 406,958 | 3.70 | 0.00 |
| | YYYY * TRJ.SO$_2$ | 7 | 1,795,535 | 256,505 | 2.33 | 0.02 |
| | YYYY * TRJ.O$_3$ | 7 | 3,997,314 | 571,045 | 5.19 | 0.00 |
| | TRJ.NO$_2$ * TRJ.O$_3$ | 1 | 3,148,433 | 3,148,433 | 28.60 | 0.00 |
| | TRJ.O$_3$ * TRJ.CO | 1 | 3,262,647 | 3,262,647 | 29.64 | 0.00 |
| | MM * TRJ.PM$_{2.5}$ | 11 | 3,280,922 | 298,266 | 2.71 | 0.00 |
| | TRJ.CO$_2$ * TRJ.TEMP | 1 | 1,584,436 | 1,584,436 | 14.39 | 0.00 |
| | TRJ.TEMP * TRJ.HUMID | 1 | 511,762 | 511,762 | 4.65 | 0.03 |
| | TRJ.WV * ShipNo | 1 | 1,976,330 | 1,976,330 | 17.95 | 0.00 |
| | TRJ.NO$_2$ * TRJ.BaP | 1 | 1,372,225 | 1,372,225 | 12.46 | 0.00 |
| | TRJ.CO * TRJ.BaP | 1 | 670,972 | 670,972 | 6.09 | 0.01 |
| | Error | 1585 | 174,489,628 | 110,088 | | |
| | Total | 2031 | 396,586,924 | | | |

**Table 4.** *Cont.*

| Dependent Variable | Effect | Degr. of Freedom | SS | MS | F | p |
|---|---|---|---|---|---|---|
| TRJ_J11 | Intercept | 1 | 1,522,477 | 1,522,477 | 318.84 | 0.00 |
| | TRJ.SO$_2$ * TRJ.HUMID | 1 | 27,526 | 27,526 | 5.76 | 0.02 |
| | TRJ.NO$_2$ * TRJ.BaP | 1 | 47,119 | 47,119 | 9.87 | 0.00 |
| | Error | 111 | 530,026 | 4775 | | |
| | Total | 113 | 577,382 | | | |
| TRJ_sum_J12_18 | Intercept | 0 | | | | |
| | MM | 7 | 8,898,402 | 1,271,200 | 24.06 | 0.00 |
| | MM * YYYY | 77 | 8,142,020 | 105,741 | 2.00 | 0.00 |
| | TRJ.NO$_2$ * TRJ.WV | 1 | 1,203,496 | 1,203,496 | 22.78 | 0.00 |
| | TRJ.PM$_{2.5}$ * TRJ.WV | 1 | 224,229 | 224,229 | 4.24 | 0.04 |
| | YYYY * TRJ.PRES | 7 | 1,510,085 | 215,726 | 4.08 | 0.00 |
| | TRJ.PM$_{2.5}$ * TRJ.BaP | 1 | 1,488,907 | 1,488,907 | 28.18 | 0.00 |
| | DD * MM | 326 | 23,524,095 | 72,160 | 1.37 | 0.00 |
| | TRJ.CO | 1 | 1,257,310 | 1,257,310 | 23.80 | 0.00 |
| | TRJ.WV * TRJ.BaP | 1 | 799,585 | 799,585 | 15.13 | 0.00 |
| | TRJ.TEMP * ShipNo | 1 | 631,362 | 631,362 | 11.95 | 0.00 |
| | TRJ.SO$_2$ * ShipNo | 1 | 487,546 | 487,546 | 9.23 | 0.00 |
| | TRJ.NO$_2$ * TRJ.HUMID | 1 | 558,340 | 558,340 | 10.57 | 0.00 |
| | TRJ.O$_3$ * TRJ.PM$_{2.5}$ | 1 | 256,889 | 256,889 | 4.86 | 0.03 |
| | Error | 1598 | 84,426,599 | 52,833 | | |
| | Total | 2028 | 158,882,411 | | | |
| TRJ_sum_j20_j22 | Intercept | 1 | 73,398,579 | 73,398,579 | 2338.56 | 0.00 |
| | TRJ.CO * TRJ.PM$_{2.5}$ | 1 | 349,231 | 349,231 | 11.13 | 0.00 |
| | YYYY * TRJ.NO | 7 | 2,089,622 | 298,517 | 9.51 | 0.00 |
| | MM | 11 | 9471,001 | 861,000 | 27.43 | 0.00 |
| | MM * YYYY | 77 | 6,273,029 | 81,468 | 2.60 | 0.00 |
| | DD * YYYY | 210 | 8,470,130 | 40,334 | 1.29 | 0.01 |
| | TRJ.CO * TRJ.BaP | 1 | 239,633 | 239,633 | 7.63 | 0.01 |
| | DD * TRJ.BaP | 30 | 1,552,638 | 51,755 | 1.65 | 0.02 |
| | TRJ.RAIN Y_N * TRJ.PM$_{2.5}$ | 1 | 261,201 | 261,201 | 8.32 | 0.00 |
| | TRJ.RAIN Y_N * TRJ.O$_3$ | 1 | 136,106 | 136,106 | 4.34 | 0.04 |
| | Error | 1603 | 50,312,033 | 31,386 | | |
| | Total | 1942 | 89,639,560 | | | |
| TRJ_J30 | Intercept | 1 | 1,009,646 | 1,009,646 | 626.11 | 0.00 |
| | YYYY * TRJ.WV | 7 | 28,776 | 4111 | 2.55 | 0.02 |
| | TRJ.RAIN Y_N * TRJ.TEMP | 1 | 17,369 | 17,369 | 10.77 | 0.00 |
| | TRJ.WV * TRJ.TEMP | 1 | 25,063 | 25,063 | 15.54 | 0.00 |
| | Error | 172 | 277,363 | 1613 | | |
| | Total | 181 | 341,021 | | | |
| TRJ_sum_J31_J34 | Intercept | 1 | 43,460,572 | 43,460,572 | 287.63 | 0.00 |
| | TRJ.WV * ShipNo | 1 | 7,373,160 | 7,373,160 | 48.80 | 0.00 |
| | TRJ.NO * TRJ.O$_3$ | 1 | 1,593,126 | 1,593,126 | 10.54 | 0.00 |
| | MM * TRJ.WV | 11 | 10,416,502 | 946,955 | 6.27 | 0.00 |
| | YYYY * TRJ.SO$_2$ | 7 | 7,325,120 | 1,046,446 | 6.93 | 0.00 |
| | TRJ.O$_3$ * TRJ.CO | 1 | 3,698,471 | 3,698,471 | 24.48 | 0.00 |
| | TRJ.NO$_2$ * TRJ.O$_3$ | 1 | 4,450,889 | 4,450,889 | 29.46 | 0.00 |
| | TRJ.PM$_{2.5}$ * ShipNo | 1 | 2,654,259 | 2,654,259 | 17.57 | 0.00 |
| | TRJ.O$_3$ * TRJ.TEMP | 1 | 1,380,829 | 1,380,829 | 9.14 | 0.00 |
| | TRJ.RAIN Y_N * TRJ.PM$_{10}$ | 1 | 1,279,038 | 1,279,038 | 8.46 | 0.00 |
| | YYYY * TRJ.NO | 7 | 3,074,650 | 439,236 | 2.91 | 0.01 |
| | Error | 1792 | 270,769,691 | 151,099 | | |
| | Total | 1824 | 303,957,358 | | | |

**Table 4.** *Cont*.

| Dependent Variable | Effect | Degr. of Freedom | SS | MS | F | *p* |
|---|---|---|---|---|---|---|
| TRJ_sum_J37_J39 | Intercept | 1 | 1,014,834 | 1,014,834 | 39.72 | 0.00 |
| | TRJ.CO$_2$ | 1 | 281,741 | 281,741 | 11.03 | 0.00 |
| | DD * TRJ.NO * TRJ.CO$_2$ | 30 | 1,222,090 | 40,736 | 1.59 | 0.02 |
| | YYYY * TRJ.PM$_{10}$ | 7 | 917,807 | 131,115 | 5.13 | 0.00 |
| | TRJ.NO * TRJ.WV | 1 | 369,943 | 369,943 | 14.48 | 0.00 |
| | YYYY * TRJ.HUMID | 7 | 1,757,588 | 251,084 | 9.83 | 0.00 |
| | Error | 1538 | 39,294,057 | 25,549 | | |
| | Total | 1584 | 44,692,024 | | | |
| TRJ_sum_J40_J42 | Intercept | 1 | 8,433,729 | 8,433,729 | 621.43 | 0.00 |
| | YYYY | 7 | 4,505,589 | 643,656 | 47.43 | 0.00 |
| | TRJ.WV * TRJ.BaP | 1 | 188,744 | 188,744 | 13.91 | 0.00 |
| | TRJ.NO$_2$ * TRJ.TEMP | 1 | 51,288 | 51,288 | 3.78 | 0.05 |
| | Error | 680 | 9,228,606 | 13,571 | | |
| | Total | 689 | 13,760,572 | | | |
| TRJ_sum_J43_J44 | Intercept | 1 | 13,123,606 | 13123606 | 991.77 | 0.00 |
| | MM | 11 | 1,001,725 | 91,066 | 6.88 | 0.00 |
| | TRJ.NO$_2$ * ShipNo | 1 | 240,792 | 240,792 | 18.20 | 0.00 |
| | YYYY * TRJ.PM$_{10}$ | 7 | 345,399 | 49,343 | 3.73 | 0.00 |
| | TRJ.O$_3$ * TRJ.BaP | 1 | 75,809 | 75,809 | 5.73 | 0.02 |
| | MM * YYYY | 77 | 1,390,801 | 18,062 | 1.37 | 0.02 |
| | YYYY * TRJ.BaP | 7 | 193,117 | 27,588 | 2.08 | 0.04 |
| | Error | 1837 | 24,308,025 | 13,232 | | |
| | Total | 1941 | 27,970,842 | | | |
| TRJ_sum_J45_J46 | TRJ.CO$_2$ * ShipNo | 1 | 2,072,385 | 2,072,385 | 22.20 | 0.00 |
| | TRJ.NO$_2$ * TRJ.WV | 1 | 1,235,310 | 1,235,310 | 13.23 | 0.00 |
| | TRJ.CO * TRJ.WV | 1 | 5,157,947 | 5,157,947 | 55.26 | 0.00 |
| | TRJ.RAIN Y_N * MM | 11 | 2,008,318 | 182,574 | 1.96 | 0.03 |
| | YYYY * TRJ.CO$_2$ | 7 | 1,659,522 | 237,075 | 2.54 | 0.01 |
| | TRJ.O$_3$ * TRJ.PM$_{10}$ | 1 | 1,366,168 | 1,366,168 | 14.64 | 0.00 |
| | TRJ.NO$_2$ | 1 | 1,248,405 | 1,248,405 | 13.37 | 0.00 |
| | TRJ.WV | 1 | 3,152,965 | 3,152,965 | 33.78 | 0.00 |
| | TRJ.NO$_2$ * ShipNo | 1 | 450,728 | 450,728 | 4.83 | 0.03 |
| | YYYY * ShipNo | 7 | 1,442,239 | 206,034 | 2.21 | 0.03 |
| | MM * TRJ.O$_3$ | 11 | 2,159,772 | 196,343 | 2.10 | 0.02 |
| TRJ_J47 | Intercept | 1 | 10,447,410 | 10,447,410 | 4522.40 | 0.00 |
| | YYYY * TRJ.WV | 7 | 66,925 | 9561 | 4.14 | 0.00 |
| | Error | 443 | 1,023,395 | 2310 | | |
| | Total | 450 | 1,090,320 | | | |
| TRJ_I20 | Intercept | 1 | 162,711,56 | 16,271,156 | 595.84 | 0.00 |
| | ShipNo | 1 | 896,973 | 896,973 | 32.85 | 0.00 |
| | YYYY * ShipNo | 7 | 1,367,431 | 195,347 | 7.15 | 0.00 |
| | TRJ.NO$_X$ * TRJ.WV | 1 | 1,119,542 | 1,119,542 | 41.00 | 0.00 |
| | TRJ.CO * TRJ.CO$_2$ | 1 | 1,574,286 | 1,574,286 | 57.65 | 0.00 |
| | TRJ.NO$_2$ * TRJ.HUMID | 1 | 558,962 | 558,962 | 20.47 | 0.00 |
| | TRJ.WV * TRJ.SO$_2$ | 1 | 309,655 | 309,655 | 11.34 | 0.00 |
| | YYYY * TRJ.CO | 7 | 580,328 | 82,904 | 3.04 | 0.00 |
| | Error | 2688 | 73,403,890 | 27,308 | | |
| | Total | 2707 | 79,599,759 | | | |

**Table 4.** *Cont.*

| Dependent Variable | Effect | Degr. of Freedom | SS | MS | F | p |
|---|---|---|---|---|---|---|
| TRJ_sum_I21_I23 | Intercept | 1 | 2,9781,054 | 29,781,054 | 1496.50 | 0.00 |
| | YYYY * TRJ.HUMID | 7 | 2,697,929 | 385,418 | 19.37 | 0.00 |
| | MM * YYYY | 77 | 3,149,772 | 40,906 | 2.06 | 0.00 |
| | TRJ.RAIN Y_N * MM | 11 | 674,651 | 61,332 | 3.08 | 0.00 |
| | TRJ.NO$_2$ * ShipNo | 1 | 213,448 | 213,448 | 10.73 | 0.00 |
| | TRJ.TEMP * TRJ.BaP | 1 | 182,901 | 182,901 | 9.19 | 0.00 |
| | TRJ.PM$_{2.5}$ * ShipNo | 1 | 154,359 | 154,359 | 7.76 | 0.01 |
| | TRJ.NO$_X$ * TRJ.WV | 1 | 72,954 | 72,954 | 3.67 | 0.06 |
| | Error | 1914 | 38,089,533 | 19,900 | | |
| | Total | 2013 | 45,278,017 | | | |
| TRJ_sum_I63_I64 | Intercept | 1 | 13,522,063 | 13,522,063 | 640.07 | 0.00 |
| | ShipNo * TRJ.CO$_2$ | 1 | 152,596 | 152,596 | 7.22 | 0.01 |
| | YYYY * TRJ.HUMID | 7 | 1,247,543 | 178,220 | 8.44 | 0.00 |
| | TRJ.NO * TRJ.TEMP | 1 | 148,988 | 148,988 | 7.05 | 0.01 |
| | TRJ.TEMP * TRJ.PM$_{2.5}$ | 1 | 195,598 | 195,598 | 9.26 | 0.00 |
| | TRJ.NO$_2$ * TRJ.O$_3$ | 1 | 148,740 | 148,740 | 7.04 | 0.01 |
| | TRJ.TEMP * TRJ.BaP | 1 | 86,417 | 86,417 | 4.09 | 0.04 |
| | Error | 2009 | 42,441,843 | 21,126 | | |
| | Total | 2021 | 44,141,351 | | | |
| TRJ_sum_I65_I66 | Intercept | 1 | 13,410,177 | 13,410,177 | 520.46 | 0.00 |
| | TRJ.TEMP * TRJ.HUMID | 1 | 586,694 | 586,694 | 22.77 | 0.00 |
| | TRJ.CO * TRJ.HUMID | 1 | 210,584 | 210,584 | 8.17 | 0.00 |
| | TRJ.RAIN Y_N * TRJ.BaP | 1 | 158,941 | 158,941 | 6.17 | 0.01 |
| | TRJ.TEMP * ShipNo | 1 | 146,044 | 146,044 | 5.67 | 0.02 |
| | Error | 1348 | 34,732,479 | 25,766 | | |
| | Total | 1352 | 35,566,024 | | | |

**Table 5.** A list of significant ($p \leq 0.05$) factors with relevant impact on disease incidence within the Agglomeration of Gdańsk [TRJ], considering the influence of sea wind (.Wsea; list of abbreviations in Appendix A); df, SS (Sum of squares), MS (Mean squares), F statistics and *p*-level of factors and interactions.

| Dependent Variable | Effect | Degr. of Freedom | SS | MS | F | p |
|---|---|---|---|---|---|---|
| TRJ_R00 | Intercept | 1 | 1027,578 | 1027,578 | 123.75 | 0.00 |
| | YYYY * TRJ.NO$_X$.Wsea | 6 | 249,503 | 41,584 | 5.01 | 0.00 |
| | DD * TRJ.O$_3$.Wsea | 30 | 447,800 | 14,927 | 1.80 | 0.01 |
| | DD * TRJ.PRES.Wsea | 30 | 416,879 | 13,896 | 1.67 | 0.02 |
| | TRJ.O$_3$.Wsea * TRJ.CO$_2$.Wsea | 1 | 40,248 | 40,248 | 4.85 | 0.03 |
| | Error | 305 | 2,532,676 | 8304 | | |
| | Total | 372 | 3,711,880 | | | |
| TRJ_R06 | Intercept | 1 | 336,1789 | 3,361,789 | 311.08 | 0.00 |
| | YYYY * TRJ.O$_3$.Wsea | 6 | 273,671 | 45,612 | 4.22 | 0.00 |
| | TRJ.CO.Wsea * TRJ.TEMP.Wsea | 1 | 67,861 | 67,861 | 6.28 | 0.01 |
| | TRJ.RAIN.Wsea Y_N * TRJ.NO.Wsea | 1 | 56,887 | 56,887 | 5.26 | 0.02 |
| | Error | 438 | 4,733,386 | 10,807 | | |
| | Total | 446 | 5,155,352 | | | |
| TRJ_R07 | Intercept | 1 | 43,369,106 | 43,369,106 | 802.24 | 0.00 |
| | YYYY * TRJ.NO$_2$.Wsea | 6 | 840,766 | 140,128 | 2.59 | 0.02 |
| | TRJ.WV.Wsea * TRJ.HUMID.Wsea | 1 | 334,803 | 334,803 | 6.19 | 0.01 |
| | YYYY * TRJ.NO$_X$.Wsea | 6 | 734,970 | 122,495 | 2.27 | 0.04 |
| | Error | 557 | 30,111,515 | 54,060 | | |
| | Total | 570 | 32,426,388 | | | |

**Table 5.** *Cont.*

| Dependent Variable | Effect | Degr. of Freedom | SS | MS | F | p |
|---|---|---|---|---|---|---|
| TRJ_sum_J00_J06 | Intercept | 0 | | | | |
| | YYYY * TRJ.PM$_{10}$.Wsea | 6 | 2082,903 | 347,150 | 3.25 | 0.00 |
| | TRJ.RAIN.Wsea Y_N * TRJ.PM$_{10}$.Wsea | 1 | 896,758 | 896,758 | 8.40 | 0.00 |
| | MM * TRJ.PRES.Wsea | 11 | 6,211,321 | 564,666 | 5.29 | 0.00 |
| | TRJ.NO$_2$.Wsea * ShipNo | 1 | 2,757,706 | 2757,706 | 25.83 | 0.00 |
| | MM * YYYY | 65 | 20,041,562 | 308,332 | 2.89 | 0.00 |
| | TRJ.NO$_2$.Wsea * TRJ.NO$_X$.Wsea | 1 | 1,923,023 | 1,923,023 | 18.01 | 0.00 |
| | TRJ.RAIN.Wsea Y_N * TRJ.BaP | 1 | 2,934,133 | 2,934,133 | 27.48 | 0.00 |
| | MM * TRJ.PM$_{10}$.Wsea | 11 | 3,983,317 | 362,120 | 3.39 | 0.00 |
| | Error | 473 | 50,506,841 | 106,780 | | |
| | Total | 570 | 116,563,428 | | | |
| TRJ_sum_J12_18 | Intercept | 1 | 15,108,160 | 15,108,160 | 306.10 | 0.00 |
| | MM * TRJ.CO$_2$.Wsea | 11 | 4,726,886 | 429,717 | 8.71 | 0.00 |
| | TRJ.WV.Wsea * TRJ.TEMP.Wsea | 1 | 344,188 | 344,188 | 6.97 | 0.01 |
| | YYYY * TRJ.BaP | 6 | 1,420,069 | 236,678 | 4.80 | 0.00 |
| | MM * TRJ.BaP | 11 | 2,099,516 | 190,865 | 3.87 | 0.00 |
| | DD * TRJ.RAIN.Wsea Y_N | 30 | 2,484,983 | 82,833 | 1.68 | 0.02 |
| | TRJ.O$_3$.Wsea * TRJ.BaP | 1 | 344,452 | 344,452 | 6.98 | 0.01 |
| | DD * TRJ.WV.Wsea | 30 | 2,726,457 | 90,882 | 1.84 | 0.00 |
| | DD * TRJ.SO$_2$.Wsea | 30 | 2,508,048 | 83,602 | 1.69 | 0.01 |
| | Error | 449 | 22,160,910 | 49,356 | | |
| TRJ_sum_j20_j22 | Intercept | 1 | 9,393,842 | 9,393,842 | 324.88 | 0.00 |
| | MM | 11 | 899,808 | 81,801 | 2.83 | 0.00 |
| | DD * TRJ.BaP | 30 | 2,313,075 | 77,102 | 2.67 | 0.00 |
| | YYYY * TRJ.NO$_X$.Wsea | 6 | 847,589 | 141,265 | 4.89 | 0.00 |
| | YYYY * TRJ.BaP | 6 | 642,652 | 107,109 | 3.70 | 0.00 |
| | ShipNo * TRJ.BaP | 1 | 223,510 | 223,510 | 7.73 | 0.01 |
| | MM * TRJ.HUMID.Wsea | 11 | 752,869 | 68,443 | 2.37 | 0.01 |
| | TRJ.NO.Wsea * TRJ.O$_3$.Wsea | 1 | 190,980 | 190,980 | 6.60 | 0.01 |
| | Error | 477 | 13,792,427 | 28,915 | | |
| | Total | 543 | 24,280,009 | | | |
| TRJ_sum_J31_J34 | Intercept | 1 | 2,283,178 | 2,283,178 | 15.24 | 0.00 |
| | TRJ.O$_3$.Wsea * TRJ.HUMID.Wsea | 1 | 2,658,824 | 2,658,824 | 17.74 | 0.00 |
| | MM * TRJ.NO.Wsea | 11 | 4,349,657 | 395,423 | 2.64 | 0.00 |
| | TRJ.WV.Wsea * ShipNo | 1 | 1,285,439 | 1,285,439 | 8.58 | 0.00 |
| | TRJ.SO$_2$.Wsea * TRJ.BaP | 1 | 939,759 | 939,759 | 6.27 | 0.01 |
| | TRJ.CO$_2$.Wsea | 1 | 828,503 | 828,503 | 5.53 | 0.02 |
| | TRJ.NO.Wsea * TRJ.NO$_2$.Wsea | 1 | 581,286 | 581,286 | 3.88 | 0.05 |
| | Error | 501 | 75,080,225 | 149,861 | | |
| | Total | 517 | 83,181,175 | | | |
| TRJ_sum_J37_J39 | Intercept | 1 | 3,075,655 | 3,075,655 | 149.96 | 0.00 |
| | TRJ.RAIN.Wsea Y_N * TRJ.BaP | 1 | 390,954 | 390,954 | 19.06 | 0.00 |
| | YYYY * TRJ.PRES.Wsea | 6 | 1,033,366 | 172,228 | 8.40 | 0.00 |
| | TRJ.HUMID.Wsea * ShipNo | 1 | 98,673 | 98,673 | 4.81 | 0.03 |
| | DD * TRJ.WV.Wsea | 30 | 934,326 | 31,144 | 1.52 | 0.04 |
| | Error | 388 | 7,957,634 | 20,509 | | |
| | Total | 426 | 10,274,790 | | | |
| TRJ_sum_J40_J42 | Intercept | 0 | | | | |
| | MM * YYYY | 59 | 566,709.4 | 9605.24 | 4.47 | 0.00 |
| | TRJ.PM$_{2.5}$.Wsea * TRJ.HUMID.Wsea | 1 | 42,343.8 | 42,343.81 | 19.70 | 0.00 |
| | TRJ.PM$_{10}$.Wsea * TRJ.PM$_{2.5}$.Wsea | 1 | 52,778.5 | 52,778.49 | 24.55 | 0.00 |
| | TRJ.RAIN.Wsea Y_N * TRJ.PRES.Wsea * ShipNo | 1 | 27,068.1 | 27,068.15 | 12.59 | 0.00 |
| | TRJ.NO$_X$.Wsea * TRJ.HUMID.Wsea | 1 | 35,865.6 | 35,865.56 | 16.68 | 0.00 |
| | TRJ.WV.Wsea * TRJ.HUMID.Wsea | 1 | 9744.2 | 9744.21 | 4.53 | 0.04 |
| | TRJ.RAIN.Wsea Y_N * TRJ.TEMP.Wsea | 1 | 32,337.0 | 32,337.02 | 15.04 | 0.00 |
| | TRJ.RAIN.Wsea Y_N * TRJ.WV.Wsea | 1 | 9,366.3 | 9366.31 | 4.36 | 0.04 |
| | Error | 86 | 184,875.7 | 2149.72 | | |
| | Total | 152 | 885,028.9 | | | |
| TRJ_sum_J43_J44 | Intercept | 1 | 3,753,231 | 3,753,231 | 252.50 | 0.00 |
| | TRJ.TEMP.Wsea | 1 | 271,379 | 271,379 | 18.26 | 0.00 |
| | TRJ.NO$_X$.Wsea * TRJ.WV.Wsea | 1 | 254,833 | 254,833 | 17.14 | 0.00 |
| | TRJ.RAIN.Wsea Y_N * TRJ.TEMP.Wsea | 1 | 60,995 | 60,995 | 4.10 | 0.04 |
| | Error | 539 | 8,011,803 | 14,864 | | |
| | Total | 542 | 8,866,679 | | | |

**Table 5.** *Cont.*

| Dependent Variable | Effect | Degr. of Freedom | SS | MS | F | p |
|---|---|---|---|---|---|---|
| TRJ_sum_J45_J46 | Intercept | 1 | 524,440 | 524,440 | 6.52 | 0.01 |
| | TRJ.WV.Wsea * ShipNo | 1 | 1,119,519 | 1,119,519 | 13.92 | 0.00 |
| | TRJ.NO$_2$.Wsea * TRJ.PRES.Wsea | 1 | 1,045,380 | 1,045,380 | 13.00 | 0.00 |
| | TRJ.PM$_{10}$.Wsea * TRJ.PM$_{2.5}$.Wsea | 1 | 487,696 | 487,696 | 6.06 | 0.01 |
| | TRJ.PRES.Wsea | 1 | 613,985 | 613,985 | 7.63 | 0.01 |
| | TRJ.RAIN.Wsea Y_N * TRJ.PM$_{2.5}$.Wsea | 1 | 406,014 | 406,014 | 5.05 | 0.03 |
| | Error | 541 | 43,517,973 | 80,440 | | |
| | Total | 546 | 46,264,315 | | | |
| TRJ_J47 | Intercept | 1 | 1,569,279 | 1,569,279 | 1260.80 | 0.00 |
| | YYYY * TRJ.BaP | 6 | 36,685 | 6114 | 4.91 | 0.00 |
| | DD * TRJ.BaP | 28 | 73,109 | 2611 | 2.10 | 0.00 |
| | YYYY * TRJ.NO.Wsea | 6 | 18,660 | 3110 | 2.50 | 0.03 |
| | Error | 87 | 108,287 | 1245 | | |
| | Total | 127 | 252,386 | | | |
| TRJ_I20 | Intercept | 1 | 16,804,098 | 16,804,098 | 561.34 | 0.00 |
| | TRJ.NO$_X$.Wsea * TRJ.WV.Wsea | 1 | 204,850 | 204,850 | 6.84 | 0.01 |
| | TRJ.RAIN.Wsea Y_N * TRJ.TEMP.Wsea | 1 | 194,479 | 194,479 | 6.50 | 0.01 |
| | Error | 538 | 16,105,371 | 29,936 | | |
| | Total | 540 | 16,442,128 | | | |
| TRJ_sum_I21_I23 | Intercept | 0 | 0 | | | |
| | TRJ.NO$_2$.Wsea * TRJ.BaP | 1 | 93,830 | 93,829.63 | 5.52 | 0.02 |
| | MM * YYYY | 65 | 1,656,422 | 25,483.42 | 1.50 | 0.01 |
| | Error | 498 | 8,467,580 | 17,003.17 | | |
| TRJ_sum_I65_I66 | Intercept | 1 | 610,483 | 610,482.7 | 24.57 | 0.00 |
| | TRJ.O$_3$.Wsea * TRJ.WV.Wsea | 1 | 457,989 | 457,988.7 | 18.43 | 0.00 |
| | TRJ.CO$_2$.Wsea * ShipNo | 1 | 265,805 | 265,804.8 | 10.70 | 0.00 |
| | Error | 369 | 9,169,514 | 24,849.6 | | |
| | Total | 371 | 9,795,077 | | | |

### 3.1. Identification of Disease Factors

The disease factors for the Agglomeration of Gdańsk were identified in two stages. The first stage was to identify models without singling out sea wind as a factor (full models), i.e., ones which considered all emission sources, both local and maritime. The second stage was to set up models for sea winds only (designation of variable: WSea) to allow the identification of factors attributable to ships entering/leaving the port and moored at berth, including those that continued to emit exhaust gases from their power generators during cargo-handling operations. The principal ingredients of pollution attributable to ships at the port include $CO_2$, CO, $SO_x$, $NO_x$ and PM [27]. With the entry into force in 2015 of the Sulphur Directive [28], the use of heavy fuel oils (HFO) was banned across the entire Baltic Sea area for ships without adequate equipment (scrubbers) to purify sulfur emissions. While shipping no longer seems to emit appreciable amounts of $SO_2$, the other pollutants, especially $NO_2$ and PM, still continue to seriously pollute urban air within the agglomeration.

With this in mind, a list was drawn up to identify diseases whose incidence within the Agglomeration of Gdańsk is attributable to air pollution generated, among other sources, by the city port. The diseases include abnormalities of heartbeat (R00), cough (R05), abnormalities of breathing (R06), pain in the throat and chest (R07), acute nasopharyngitis, acute sinusitis, acute pharyngitis, acute tonsilitis, acute laryngitis and tracheitis, acute obstructive laryngitis and epiglottitis, acute upper respiratory infections of multiple or unspecified sites (J00–J06), influenza due to unidentified virus (J11), viral pneumonia (J12–J18), acute bronchitis, broncholitis and unspecified acute lower respiratory infection (J20–J22), vasomotor and allergic rhinitis (J30), chronic rhinitis, nasopharyngitis and pharyngitis, sinusitis, nasal polyp and other disorders of the nose and nasal sinuses (J31–J34), chronic laryngitis and laryngotracheitis, diseases of the vocal cords and larynx not elsewhere classified, and other diseases of the upper respiratory tract (J37–J39), bronchitis (J40–J42), emphysema and other chronic obstructive pulmonary diseases (J43–J44), asthma and status asthmaticus (J45–J46), bronchiectasis (J47), angina pectoris (I20), acute myocardial infarction, subsequent myocardial infarction

and certain current complications following acute myocardial infarction (I21–I23), cerebral infarction and stroke not specified as haemorrhage or infarction (I63–I64), as well as occlusion and stenosis of precerebral arteries not resulting in cerebral infarction (I65–I66). However, their correlation factor (compare the "%Var" columns in Tables 2 and 3) ranges widely from 2.0% to 79.1%. Also, separate analyses are required for outcomes obtained for urban pollution not including shipping operations at port and for pollution outcomes including that factor. The details are presented in Tables 2 and 3.

The most noteworthy three out of the 19 identified diseases are acute severe asthma, chronic obstructive pulmonary disease and pneumonia. Air pollution penetrates into the body by means of the lungs, which act as a protective filter and therefore bear the brunt of infections trasmitted by air. The most severe consequences include pneumonia, which is a cause of numerous deaths, acute severe asthma, as well as COPD, a life-threatening condition which exposes the healthcare system in Poland to considerable financial losses. For that reason, the further discussion will focus in detail on these three diseases to explain the correlations indentified by stochastic models.

### 3.1.1. Acute severe asthma [J45–J46]

The applied full models account for 12.2% of variability in incidence rates for bronchial asthma ($R^2$) depending on air pollution. Statistically relevant factors correlating with incidence include:

$CO_2$ * ShipNo | $NO_2$ * WV | CO*WV | RAIN * MM | YYYY*$CO_2$ | $O_3$ * $PM_{10}$ | $NO_2$ | WV | $NO_2$ * ShipNo | YYYY * ShipNo | MM*$O_3$

The findings point to a strong influence of annual seasonality with periodical peaks of incidence related to natural trends, variable concentrations of $CO_2$ and $O_3$, and changing patterns of ship traffic where causes may include the spread of chemical compounds resulting from loading/unloading operations ($CO_2$, $NO_2$).

The findings point to the prominent role of $CO_2$, $O_3$ and $NO_2$. These compounds are powerful on their own ($NO_2$), as well as in combination with wind (WV). Interestingly, a statistically relevant interaction of $O_3$ with particulate matter $PM_{10}$ has been found.

Models for sea wind direction account for approx. 6% of variability in incidence rates for asthma. The following variables have been identified as statistically relevant factors:

WV.Wsea * ShipNo | $NO_2$.Wsea * PRES.Wsea | $PM_{10}$.Wsea * $PM_{2.5}$.Wsea | PRES.Wsea | RAIN.Wsea * $PM_{2.5}$.Wsea

This is a refinement of the findings explaining the influence of pollutants ($NO_2$, $PM_{10}$) attributable to ship traffic, and of $PM_{2.5}$ in conjunction with PM10 and in combination with rain.

### 3.1.2. Chronic obstructive pulmonary disease (COPD) [J43–J44]

Full models account for 13.1% of variability incidence rates for COPD (cf. Table 2). Statistically relevant factors correlating with incidence include:

MM | TRJ.$NO_2$ * ShipNo | YYYY*TRJ.PM10 | TRJ.$O_3$ * TRJ.BaP

The findings show that the factors with the most impact on incidence of COPD within the Tricity area include: $NO_2$ related to ship traffic, $PM_{10}$ and interactions between $O_3$ and BaP.

Another factor with a strong influence on incidence is seasonality. Models for sea wind direction account for approx. 9.6% of the variability. Statistically relevant factors include the following variables:

TEMP.Wsea | NOX.Wsea*WV.Wsea | RAIN.Wsea Y_N*TEMP.Wsea

That is to say, temperature fluctuations and seasonality of incidence as well as NOx in conjunction with sea wind (WV.Wsea).

### 3.1.3. Pneumonia [TRJ_sum_J12_18]

Full models account for 46.9% of variability in incidence rates for pneumonia (Table 2). This is among the highest values, pointing to a strong correlation with incidence.

Statistically relevant factors include the following variables:

MM | MM * YYYY | $NO_2$ * WV | $PM_{2.5}$ * WV | YYYY * PRES | $PM_{2.5}$ * BaP | DD * MM | CO. WV * BaP | TEMP * ShipNo | $SO_2$ * ShipNo | $NO_2$ * HUMID | $O_3$ * $PM_{2.5}$

The findings show that the leading factors with impact on the incidence of pneumonia within the Tricity area include: $NO_2$, $PM_{2.5}$, CO, BaP and $SO_2$ related to emissions from ships.

In addition to the seasonality (MM.YYYY.DD), incidence is also affected by weather conditions (wind, temperature, humidity). Noteworthy in this regard are interactions between $O_3$ and $PM_{2.5}$ and between ship traffic and temperature (TEMP * ShipNo).

Models for sea wind direction account for approx. 49% of the variability, a high proportion. Statistically relevant factors include the following variables:

MM * $CO_2$.Wsea | WV.Wsea * TEMP.Wsea | YYYY * BaP | MM * BaP | DD * RAIN.Wsea Y_N | $O_3$.Wsea * BaP | DD * WV.Wsea | DD * $SO_2$.Wsea

These models give a more detailed picture of the situation by pointing to a strong influence of BaP, both periodically (YYYY * BaP) and in conjunction with $O_3$. Also, previous findings have been confirmed: $CO_2$ and $SO_2$ blown in from the sea and ports have a serious impact on incidence.

### 3.2. Identification of the Financial Costs of Medical Treatment

Treatment costs to medical service providers have been calculated for the 19 diseases identified above on the basis of medical treatment price lists for various types of respiratory diseases supplied by the Military Institute of Medicine. Due consideration has been given to the criterion of relevance to life safety. The calculation covers only three diseases named in Section 3.1, items 1–3. This has helped to establish the average cost rates for each of the five cost groups and the amounts refunded by the National Health Fund. The figures are set out in Table 6. Even a cursory analysis has shown a wide disparity between actual costs paid by medical service providers (hospitals) and amounts recovered from the National Health Fund in reimbursement for these services. Funding gaps are widest for pneumonia. The figures presented below may be utilized in the future to conduct a more detailed study of the Polish healthcare system with the aim of closing the gap between medical costs and available refunds. This also shows the scale of expenditure on specific diseases.

**Table 6.** Financial costs of medical treatment of selected diseases related to air pollution within the Agglomeration of Gdańsk in 2018 (1 euro = PLN 4.3).

| Disease | Acute Severe Bronchial Asthma J45–J46 | Chronic Obstructive Pulmonary Disease (COPD) J43–J44 | Pneumonia J12–J18 |
|---|---|---|---|
| Avg. length of hospitalization | 8.64 days | 6.4 days | 13.41 days |
| Avg. Polish National Health Fund (NFZ) refund | PLN 4179.36 | PLN 2243.47 | PLN 3191.06 |
| Avg. medical care cost | PLN 4076.19 | PLN 2811.82 | PLN 6239.33 |
| Avg. daily hospitalization cost | PLN 1259.90 | PLN 899.31 | PLN 1935.67 |
| Avg. diagnostic cost | PLN 597.82 | PLN 401.33 | PLN 1312.11 |
| Avg. medication cost | PLN 255.86 | PLN 99.84 | PLN 929.31 |
| Avg. consultation cost | PLN 36.00 | PLN 19.20 | PLN 69.88 |
| Total of average costs | PLN 6225.77 | PLN 4231.50 | PLN 10,486.30 |
| Average financial result | PLN −2046.41 | PLN −1988.04 | PLN −7295.25 |
| Number of incidents | 1318 | 851 | 2100 |
| Total cost of treatment | PLN 8,205,563.86 | PLN 3,601,006.50 | PLN 22,021,230.00 |
| Total cost refunded | PLN 5,508,396.48 | PLN 1,909,192.97 | PLN 6,701,226.00 |
| Funding gap | PLN −2,695,167.38 | PLN −1,691,813.53 | PLN −15,320,004.00 |

## 4. Discussion

### 4.1. Regarding the Health and Environmental Component

The research presented in this paper is a preliminary pilot study bringing together, for the first time, a vast collection of over 14 million records on medical services with environmental data. The study has clearly shown a cause-and-effect relationship between air pollution generated by industrial operations, ports and shipping, and disease incidence rates within the Agglomeration of Gdańsk. However, the results are far from final. More research is necessary to study other agglomerations and cities for a more accurate understanding of relations between the impact factors, the agglomeration's location and the associated pollution levels. Such further research will verify the accuracy of the GRM model.

Another area of concern relates to the coverage of data collected by public health institutions (mainly the National Health Fund—NFZ) and environmental monitoring bodies. Record-keeping should preferably evolve towards a detailed description of medical services to allow the identification of the gender and age of patients, length of medical leave due to a particular disease, and type of medical services provided. As for environmental data, it would be desirable to increase the number of atmospheric measurement stations to allow the investigation of the spatial distribution of incidence of diseases attributable to air pollution. It would also be useful to map the location of the major emitters of pollutants (e.g., refineries, garbage incineration plants and timber mills).

The areas for future research into the impact of pollution on diseases include both methodological and factual aspects of the problem. Further work in these areas should identify pollutants with the greatest impact on the incidence of various diseases in society and, by extension, also set up a framework to combat pollution. Such a framework should indicate action to target those sources of pollution whose elimination will be the most beneficial.

### 4.2. Regarding the Economic Component

Within the economic section of the paper, some concerns may arise as to the calculated average costs of treatment of selected diseases. However, without listing these costs separately by specific types of medical services (hospitalization, medication, diagnostics, etc.), it would be impossible to accurately identify which financial costs are incurred by urban air pollution. That is why it is necessary to keep records listing the costs incurred by a medical service provider in offering specific medical procedures. Such records should preferably indicate the amount of National Health Fund (NFZ) refunds provided under contract with a specific medical service provider. Such a step will paint a more accurate picture of the Polish healthcare system's funding shortfall.

Also, establishing a correlation between incidence rates for diseases resulting from air pollution with their treatment costs would allow for a much more adequate framework to be set up to support the process of treatment and to match refunds with service provider needs. In effect, this would considerably reduce the disparities between financial expectations and actual cash flows. It would therefore be necessary to examine in more detail the issue of shortfall in the funding.

With respect to further detailed research, it would be desirable to analyze the cost variability over time of the treatment of diseases related to ship traffic within ports so as to identify the economic impact of ports on the health condition of populations residing in port cities. It would be especially interesting to look at cost variation in the Baltic Sea Region in response to the 2015 Sulphur Directive.

A further step with interesting implications for social economics would be to extend the calculations to the external costs of diseases. These costs would have to be correlated with average salaries, the resulting lost earnings, costs to employers, who—according to Polish law—are obliged to pay sickness benefits for the first 33 days of sick leave, and costs to the Polish Social Insurance Company (ZUS), which is obliged to provide sick pay from the 34th day onwards. This would set the stage for even more in-depth research to identify the full indirect costs of diseases and total economic costs of urban air pollution. This would have the added benefit of elaborating in-depth cost optimization models

for medical treatment, while also identifying extreme deviations from average values, the structure of medical treatment costs, and how these figures differ across various medical service recipients.

## 5. Conclusions

This study is a pilot project, as the identified models require in-depth analysis and in-depth interpretation in order to fully understand the impact on every disease. It is also necessary to carry out a comparative study of the findings with their counterparts for other Polish cities such as Cracow and Warsaw. Whatever findings are now available represent high value as they reveal, for the first time in Poland, some significant impact factors and their interactions that contribute to selected diseases in the long term. However, further research is necessary to fully understand the statistical importance of disease factors depending on the region and degree of local pollution. This would allow for a full estimate of costs to urban populations as a result of industrial and urban air pollution.

**Author Contributions:** P.O.C.—draft preparation, quantitative analysis, software, data validation, modeling, writing; P.D.—draft preparation, medical analysis, data, writing; A.O.-J.—draft preparation, economic analysis, validation, writing; M.B.—environmental analysis, obtaining data; E.C.—draft preparation, economic analysis, validation, writing, final editing; T.O.—quantitative analysis, modeling; P.R.-K.—environmental analysis, obtaining data; A.B.—environmental analysis, obtaining data, writing, final editing. All authors have read and agreed to the published version of the manuscript.

**Funding:** This research received no external funding.

**Conflicts of Interest:** The authors declare no conflict of interest.

## Appendix A

**Table A1.** List of all variables used for cause-and-effect models.

| Reference Automatic Measurement Results (1 h; $\mu g/m^3$) Aggregated (Averages) to a Day in Models | |
|---|---|
| **Variable** | **Description** |
| **TRJ.SO$_2$** [SO$_2$ in tables] | SO$_2$ [$\mu g/m^3$] |
| **TRJ.NO** | NO [$\mu g/m^3$] |
| **TRJ.NO$_2$** [NO$_2$ in tables] | NO$_2$ [$\mu g/m^3$] |
| **TRJ.NO$_X$** [NOx in tables] | NO$_X$ [$\mu g/m^3$] |
| **TRJ.O$_3$** [O$_3$ in tables] | O$_3$ [$\mu g/m^3$] |
| **TRJ.CO** | CO [$\mu g/m^3$] |
| **TRJ.CO$_2$** [CO$_2$ in tables] | CO$_2$ [$\mu g/m^3$] |
| **TRJ.PM$_{10}$** [PM$_{10}$ in tables] | PM$_{10}$ [$\mu g/m^3$] |
| **TRJ.PM$_{2.5}$** [PM$_{2.5}$ in tables] | PM$_{2.5}$ [$\mu g/m^3$] |
| **TRJ.BaP** | Benzapirene [$\mu g/m^3$] |
| **TRJ.PRES** | Atmospheric pressure [hPa] |
| **TRJ.WV** | Wind speed [m/s] |
| **TRJ.TEMP** | Temperature [degrees Celsius] |
| **TRJ.HUMID** | Humidity [%] |
| **TRJ.RAIN** | Rainfall [mm] |
| **TRJ.SO$_2$.Wsea** | SO$_2$ [$\mu g/m^3$] |
| **TRJ.NO.Wsea** | NO [$\mu g/m^3$] |
| **TRJ.NO$_2$.Wsea** | NO$_2$ [$\mu g/m^3$] |
| **TRJ.NO$_X$.Wsea** | NO$_X$ [$\mu g/m^3$] |
| **TRJ.O$_3$.Wsea** | O$_3$ [$\mu g/m^3$] |
| **TRJ.CO.Wsea** | CO [$\mu g/m^3$] |
| **TRJ.CO$_2$.Wsea** | CO$_2$ [$\mu g/m^3$] |
| **TRJ.PM$_{10}$.Wsea** | PM$_{10}$ [$\mu g/m^3$] |
| **TRJ.PM$_{2.5}$.Wsea** | PM$_{2.5}$ [$\mu g/m^3$] |
| **TRJ.PRES.Wsea** | Atmospheric pressure [hPa] |
| **TRJ.WV.Wsea** | Wind speed [m/s] |
| **TRJ.TEMP.Wsea** | Temperature [degrees Celsius] |
| **TRJ.HUMID.Wsea** | Humidity [%] |
| **TRJ.RAIN.Wsea** | Rainfall [mm] |

**Table A1.** *Cont.*

| Number of provisions aggregated (sum) to a day in models | |
|---|---|
| **ICD10 Code** | **Description** |
| TRJ_I20 | Coronary artery disease |
| TRJ_I21 | Acute heart attack |
| TRJ_I22 | Another heart attack (reinfarction) |
| TRJ_I23 | Complications occurring during acute myocardial infarction |
| TRJ_I24 | Other acute forms of ischemic heart disease |
| TRJ_I25 | Chronic ischemic heart disease |
| TRJ_I46 | Cardiac arrest |
| TRJ_I47 | Paroxysmal tachycardia |
| TRJ_I48 | Atrial fibrillation |
| TRJ_I49 | Other cardiac arrhythmia |
| TRJ_I50 | Heart failure |
| TRJ_I51 | Heart disease not precisely defined and complications of heart disease |
| TRJ_I52 | Other cardiac dysfunction in diseases classified elsewhere |
| TRJ_I63 | Cerebral infarction |
| TRJ_I64 | Stroke, not defined as hemorrhagic or infarcted |
| TRJ_I65 | Blockage and narrowing of the pre-cerebral arteries that do not cause cerebral infarction |
| TRJ_I66 | Blockage and narrowing of the cerebral arteries that do not cause cerebral infarction |
| TRJ_I67 | Other cerebrovascular diseases |
| TRJ_I68 | Cerebrovascular disorders in diseases occurring elsewhere |
| TRJ_I69 | Consequences of cerebrovascular diseases |
| TRJ_R00 | Heart disorders |
| TRJ_R05 | Cough |
| TRJ_R06 | Breathing disorders |
| TRJ_R07 | Sore throat and chest |
| TRJ_J00 | Acute inflammation of the nose and throat (common cold) |
| TRJ_J01 | Acute sinusitis |
| TRJ_J02 | Acute pharyngitis |
| TRJ_J03 | Acute tonsillitis |
| TRJ_J04 | Acute laryngotracheitis |
| TRJ_J05 | Acute obstructive laryngitis and epiglottitis |
| TRJ_J06 | Acute upper respiratory tract infection with multiple or unspecified localization |
| TRJ_J11 | Flu caused by an unidentified virus |
| TRJ_J12 | Viral pneumonia, not elsewhere classified |
| TRJ_J13 | Streptococcal pneumonia (*Streptococcus pneumoniae*) |
| TRJ_J14 | Pneumonia caused by influenza bacillus (*Haemophilus influenzae*) |
| TRJ_J15 | Bacterial pneumonia, not elsewhere classified |
| TRJ_J16 | Pneumonia caused by other microorganisms not elsewhere classified |
| TRJ_J17 | Pneumonia in diseases classified elsewhere |
| TRJ_J18 | Pneumonia caused by an unspecified microorganism |
| TRJ_J20 | Acute bronchitis |
| TRJ_J21 | Acute bronchiolitis |
| TRJ_J22 | Unspecified acute lower respiratory infection |
| TRJ_J30 | Angioedema and allergic rhinitis |
| TRJ_J31 | Chronic nasopharyngitis |
| TRJ_J32 | Chronic sinusitis |
| TRJ_J33 | Nasal polyp |
| TRJ_J34 | Other diseases of the nose and paranasal sinuses |
| TRJ_J35 | Chronic tonsil and pharyngeal tonsil diseases |
| TRJ_J36 | Peritonsillar abscess |
| TRJ_J37 | Chronic laryngitis and tracheitis |
| TRJ_J38 | Inflammation of the vocal cords and larynx, not elsewhere classified |
| TRJ_J39 | Other diseases of the upper respiratory tract |
| TRJ_J40 | Bronchitis not defined as acute or chronic |
| TRJ_J41 | Chronic, simple, and mucopurulent bronchitis |
| TRJ_J42 | Unspecified chronic bronchitis |
| TRJ_J43 | Emphysema |
| TRJ_J44 | Other chronic obstructive pulmonary disease |
| TRJ_J45 | Bronchial asthma |
| TRJ_J46 | Status asthmaticus |
| TRJ_J47 | Bronchiectasis |

**Table A1.** *Cont.*

| Time variables (binary in models) and ship numbers | |
| --- | --- |
| **Variable** | **Description** |
| **DD** | Day |
| **MM** | Month |
| **YYYY** | Year |
| **ShipNo** | Number of ships entering the port of Gdańsk |
| **Abbreviations:** | |
| TRJ | Tricity Agglomeration (Gdańsk) |
| AMxx | Designation of ARMAAG measuring stations |
| Wsea | Measurement results for winds blowing from the sea |

**Appendix B**

Due to the volume (several thousand verified factor interactions), the final results of significance tests will be presented synthetically.

The GRM model identification process took place in three main general stages:

In the first stage, models without interaction were tested, and the significance of each parameter was tested using standard tests based on the Student's t distribution, the significance of the model based on Fisher's distribution (F test), and the degree of variation explanation by the model (multiple R, $R^2$ and corrected $R^2$).

Second, the next step was to identify, in addition, all statistically significant interactions of independent variables to the second degree. For this purpose, two parallel iterative procedures for model building were used: forward stepwise and best subset with the $R^2$ criterion. As a result of comparing the results of both iterative procedures, independent variables significantly related to the dependent variable (selected cases of diseases) were selected.

The final stage of model identification was to build the model only with variables statistically significantly associated with the dependent variable. If there was more than one such model, the one for which $R^2$ was higher was chosen and the assumptions of its applicability were examined, i.e., the normality of the random component within the level of each factor, and each interaction, was assessed separately. Due to the large volume of results for presentation, a table was built with the names of factors significantly related to the dependent variable (according to Appendix A).

In some cases, it was necessary to repeatedly test various factor systems and their interactions, despite the use of iterative procedures. The final model was a model with the least number of factors at the highest or similar level of explained variance to more complex models.

In the intermediate identification stages, selected additional tools were used: Pareto charts, Ljung Box Pierce Q tests (in model stationary testing), and another visualization tool.

Appendix B presents the final working result (print from the Statistica system, with manual corrections) tables of model estimations, results of significance tests for each factor and interaction in the models, as well as selected intermediate stages of the process of identifying selected variables. Due to the size of the result sets (in the order of several thousand pages), it is not possible to present all the detailed result sets.

Sample, selected results of the model identification process for TRJ_sum_J00_J06:

| **Model Without Interactions:** | **Effect** | **Sum of Squares (SS)** | **df; degr. ff freedom** | **Mean Squares (MS)** | **F** | ***p*** |
|---|---|---|---|---|---|---|
| **TRJ_sum_J00_J06** | **Univariate Tests of Significance for TRJ_sum_J00_J06 (#TRJ DD2010_2018 in NFZ main 2010 WORK EN v093.stw)** | | | | | |
| | **Sigma-Restricted Parameterization** | | | | | |
| | **Effective Hypothesis Decomposition; Std. Error of Estimate: 360,2027** | | | | | |
| | **Include Condition: YYYY ≥ 2010 AND YYYY ≤ 2018** | | | | | |
| | Intercept | 694,149 | 1 | 694,149 | 5.35006 | 0.020824 |
| | $TRJ.SO_2$ | 717,53 | 1 | 717,53 | 0.55303 | 0.457171 |
| | TRJ.NO | 574,957 | 1 | 574,957 | 4.43141 | 0.035410 |
| | $TRJ.NO_2$ | 478,351 | 1 | 478,351 | 3.68683 | 0.054988 |
| | $TRJ.NO_X$ | 599,354 | 1 | 599,354 | 4.61944 | 0.031733 |
| | $TRJ.O_3$ | 11,901 | 1 | 11,901 | 0.09172 | 0.762030 |
| | TRJ.CO | 1,013,120 | 1 | 1,013,120 | 7.80848 | 0.005251 |
| | $TRJ.CO_2$ | 4433 | 1 | 4433 | 0.03417 | 0.853365 |
| | $TRJ.PM_{10}$ | 470,499 | 1 | 470,499 | 3.62631 | 0.057019 |
| | $TRJ.PM_{2.5}$ | 2,297,590 | 1 | 2,297,590 | 17.70837 | 0.000027 |
| | TRJ.PRES | 198,225 | 1 | 198,225 | 1.52779 | 0.216592 |
| | TRJ.WV | 1,978,574 | 1 | 1,978,574 | 15.24960 | 0.000097 |
| | TRJ.TEMP | 1,170,582 | 1 | 1,170,582 | 9.02210 | 0.002701 |
| | TRJ.HUMID | 427,233 | 1 | 427,233 | 3.29284 | 0.069735 |
| | ShipNo | 684,749 | 1 | 684,749 | 5.27762 | 0.021706 |
| | TRJ.BaP | 43,494 | 1 | 43,494 | 0.33523 | 0.562664 |
| | DD | 2,309,906 | 30 | 76,997 | 0.59344 | 0.960926 |
| | MM | 43,002,543 | 11 | 3,909,322 | 30.13058 | 0.000000 |
| | YYYY | 13,655,564 | 7 | 1,950,795 | 15.03549 | 0.000000 |
| | TRJ.RAIN Y_N | 78,129 | 1 | 78,129 | 0.60217 | 0.437844 |
| | Error | 255,210,364 | 1967 | 129,746 | | |

| **Dependent Variable** | **Test of SS Whole Model vs. SS Residual (#TRJ DD2010_2018 in NFZ main 2010 WORK EN v093.stw) Include Condition: YYYY ≥ 2010 AND YYYY ≤ 2018** | | | | | | | | | | |
|---|---|---|---|---|---|---|---|---|---|---|---|
| **Effect** | **Multiple R** | **Multiple $R^2$** | **Adjusted $R^2$** | **SS** | **df** | **MS** | **SS** | **df** | **MS** | **F** | ***p*** |
| TRJ_sum_J00_J06 | 0.60 | 0.36 | 0.34 | 141376559 | 64 | 2,209,009 | 255,210,364 | 1967 | 129,746.0 | 17.02564 | 0.00 |

| **9** | **Effect** | **Summary of Stepwise Regression; Variable: TRJ_sum_J00_J06 (#TRJ DD2010_2018 in NFZ main 2010 WORK EN v093.stw)** | | | | | |
|---|---|---|---|---|---|---|---|
| | | **Forward Stepwise P to Enter: 0.05; P to remove: 0.05** | | | | | |
| | | **Include Condition: YYYY ≥ 2010 AND YYYY ≤ 2018** | | | | | |
| | **Effect** | **Steps** | **Degr. Of Freedom** | **F to Remove** | **P to Remove** | **F to Enter** | **P to Enter** | **Effect Status** |
| | MM | Step Number 24 | 10 | 12.008 | 0.000 | | | In |
| | $TRJ.CO_2$ * TRJ.TEMP | | 1 | 14.356 | 0.000 | | | In |
| | DD * MM | | 330 | 1.269 | 0.002 | | | In |
| | $TRJ.NO_2$ * $TRJ.O_3$ | | 1 | 28.527 | 0.000 | | | In |
| | TRJ.WV * ShipNo | | 1 | 17.907 | 0.000 | | | In |
| | YYYY * $TRJ.O_3$ | | 7 | 5.174 | 0.000 | | | In |
| | MM * YYYY | | 77 | 3.687 | 0.000 | | | In |
| | $TRJ.O_3$ * TRJ.CO | | 1 | 29.562 | 0.000 | | | In |
| | MM * $TRJ.PM_{2.5}$ | | 11 | 2.702 | 0.002 | | | In |
| | $TRJ.NO_2$ * TRJ.BaP | | 1 | 12.433 | 0.000 | | | In |
| | TRJ.CO * TRJ.BaP | | 1 | 6.079 | 0.014 | | | In |
| | YYYY * $TRJ.SO_2$ | | 7 | 2.324 | 0.023 | | | In |
| | TRJ.TEMP * TRJ.HUMID | | 1 | 4.637 | 0.031 | | | In |

| Dependent Variable | Test of SS Whole Model vs. SS Residual (#TRJ DD2010_2018 in NFZ main 2010 WORK EN v093.stw) Include Condition: YYYY ≥ 2010 AND YYYY ≤ 2018 | | | | | | | | | | |
|---|---|---|---|---|---|---|---|---|---|---|---|
| Effect | Multiple R | Multiple $R^2$ | Adjusted $R^2$ | SS | df | MS | SS | df | MS | F | *p* |
| TRJ_sum_J00_J06 | 0.75 | 0.56 | 0.44 | 22,209,729 | 446 | 497,976 | 174,489,628 | 1585 | 110,088 | 4.52 | 0.00 |

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
