# Peer review of "A Preliminary Attempt at the Identification and Financial Estimation of the Negative Health Effects of Urban and Industrial Air Pollution Based on the Agglomeration of Gdańsk"

_sustainability, doi:10.3390/su12010042_

Round 1

Reviewer 1 Report

In this paper the authors determined the impact of air pollutants as well as of seasonality, land- and sea-based emissions and their mutual interactions on long-term selected health outcomes in terms of increased incidence and direct costs of medical treatment in Agglomeration of Gdańsk in Poland in the years 2010-2018 by Generalized Regression Models. The authors identified the disease factors first considering all emission sources, then the factors attributable to emissions of maritime and port origin, and at the same time a list of diseases whose incidence may be associated to air pollution. The results detected three diseases whose incidence is statistically influenced by environmental factors and for these diseases financial costs were computed. The manuscript is well written and of interest, providing important insights on the relationship between exposure to air pollutants and incidence of diseases.

Specific comments below are in order of appearance:

In Table 1 the authors should add results from the normality test.

Both in Table 2 and Table 3 a significance test is recommended.

Table 3. The title seems identical to that of Table 2. Moreover, what is the meaning of PRES? A list of abbreviations below the tables could be useful.

Line 281. Please correct the typological error in “sulphur”.

Line 321. I do not find the interaction between NO2 and rain among the statistically relevant factors.

Author Response

First of all, we would like to thank you for in-depth reviews that have helped us significantly improve the description of our work results. Valuable and detailed reviews.

We refer to some of the comments descriptively in the responses to the comments. Most comments have been taken into account in the form of changes directly in the content of the article.

Reviewer 1

Review Report Form 1

Open Review

English language and style

( ) Extensive editing of English language and style required 
( ) Moderate English changes required 
(x) English language and style are fine/minor spell check required 
( ) I don't feel qualified to judge about the English language and style 

Yes

Can be improved

Must be improved

Not applicable

Does the introduction provide sufficient background and include all relevant references?

(x)

( )

( )

( )

Is the research design appropriate?

(x)

( )

( )

( )

Are the methods adequately described?

(x)

( )

( )

( )

Are the results clearly presented?

( )

(x)

( )

( )

Are the conclusions supported by the results?

(x)

( )

( )

( )

Comments and Suggestions for Authors

In this paper the authors determined the impact of air pollutants as well as of seasonality, land- and sea-based emissions and their mutual interactions on long-term selected health outcomes in terms of increased incidence and direct costs of medical treatment in Agglomeration of Gdańsk in Poland in the years 2010-2018 by Generalized Regression Models. The authors identified the disease factors first considering all emission sources, then the factors attributable to emissions of maritime and port origin, and at the same time a list of diseases whose incidence may be associated to air pollution. The results detected three diseases whose incidence is statistically influenced by environmental factors and for these diseases financial costs were computed. The manuscript is well written and of interest, providing important insights on the relationship between exposure to air pollutants and incidence of diseases.

Specific comments below are in order of appearance:

In Table 1 the authors should add results from the normality test.

The technical details of the response are contained in the attached change log file.

The results of normality tests (KS, Lil., SW) were included and, in addition, estimation by the most reliable method was performed to identify similar distributions. The results of the work are included in the revised article and the comments.

Both in Table 2 and Table 3 a significance test is recommended.

Table 2 and Table 3 have been modified to test the significance of each model (columns F and p). Appendix B and tables 4 and table 5 with tests for each factor and interaction in the models were added.

Table 3. The title seems identical to that of Table 2. Moreover, what is the meaning of PRES? A list of abbreviations below the tables could be useful.

The description of variables has been corrected in Appendix A. Table 2 shows the results for full models and Table 3 for models with winds blowing from the sea.

Line 281. Please correct the typological error in “sulphur”.

Fixed.

Line 321. I do not find the interaction between NO2 and rain among the statistically relevant factors.

Corrected. There is no such interaction.

Submission Date

16 November 2019

Date of this review

21 Nov 2019 10:12:38

Submission Date

16 November 2019

Date of this review

28 Nov 2019 04:10:51

Response:

5.12.2019

5 Dec 2019

Reviewer 2 Report

This manuscript describes the economic impact of pollution in the Gdansk region of Poland. The manuscript's theme is of great interest as the monetary consequence of pollution is a critical contribution. The structure is neither standard nor divided into subsections which makes the long sections fairly difficult to read. There are several spelling and grammar errors that should be corrected before publishing.

Specific Comments:

The introduction section should be rewritten for increased readibility. As it is a fairly long section, it would help if sub-sections were used (i.e. 1.1. Motivation, 1.2 Literature Review, etc.) since these components are not clearly defined. Line 88, a comma "," is used instead of a period "." as a decimal marker. It is unclear what the pollution measurements are in relation to the statistical analysis. The authors need to discuss this in more detail. Are these annual values, peak hourly values, averages across all stations, etc.? This must be further developed. Figure 1 does not have the standard caption format. The colors are too faint to distinguish among them. The legend, as well as city names on the map, are too small to be readable. Since most readers may not be familiar with the study region, including latitude/longitude coordinates would be a welcome addition. Line 181 seems to have an extra indentation. The method section could be divided into "Data", "Statistical Analysis", etc. for increased readability. Line 183 uses a symbol that does not look like "Y" in the equation. The results section is very difficult to follow. The first part of the results seems very disorganized with changing bolded/unbolded text. While the Appendix is referred to in the Abstract (which should not be the case), it is not referred to before the tables in the result section. This is necessary because the variables are only defined in the Appendix and otherwise the reader cannot follow the results. In Table 1, the authors should consider listing the "Valid N" values together with a comma (no space in the thousands/hundreds) because it makes it difficult to read. Line 281 has "sulphur" /"sulfur" misspelled. Line 295 has "obstructive" misspelled. Line 343 and previous instances have a confusing grey font which makes it difficult to read. I am not sure it is necessary to state that the source of data is "Author's own work" as that is implied. If it is not, that should be cited, but otherwise it is assumed the authors provided the data. In Table 4, row 2, what is the cost unit for Pneumonia? Lines 376 and 397 should be subsections for clarity. It seems that the authors have repeated information from the Discussion in the Conclusions. For example future work should be better detailed and separated.

Author Response

First of all, we would like to thank you for in-depth reviews that have helped us significantly improve the description of our work results. Valuable and detailed reviews.

We refer to some of the comments descriptively in the responses to the comments. Most comments have been taken into account in the form of changes directly in the content of the article.

Reviewer  2

Review Report Form

Open Review

English language and style

( ) Extensive editing of English language and style required 
(x) Moderate English changes required 
( ) English language and style are fine/minor spell check required 
( ) I don't feel qualified to judge about the English language and style 

Yes

Can be improved

Must be improved

Not applicable

Does the introduction provide sufficient background and include all relevant references?

( )

( )

(x)

( )

Is the research design appropriate?

( )

(x)

( )

( )

Are the methods adequately described?

( )

(x)

( )

( )

Are the results clearly presented?

( )

( )

(x)

( )

Are the conclusions supported by the results?

( )

(x)

( )

( )

Comments and Suggestions for Authors

This manuscript describes the economic impact of pollution in the Gdansk region of Poland. The manuscript's theme is of great interest as the monetary consequence of pollution is a critical contribution. The structure is neither standard nor divided into subsections which makes the long sections fairly difficult to read. There are several spelling and grammar errors that should be corrected before publishing.

Specific Comments:

The introduction section should be rewritten for increased readibility. As it is a fairly long section, it would help if sub-sections were used (i.e. 1.1. Motivation, 1.2 Literature Review, etc.) since these components are not clearly defined.

Corrected. The modified text is in the new version of the article.

 Line 88, a comma "," is used instead of a period "." as a decimal marker. It is unclear what the pollution measurements are in relation to the statistical analysis.

Corrected.

 The authors need to discuss this in more detail. Are these annual values, peak hourly values, averages across all stations, etc.? This must be further developed.

Corrected. The modified text is in the new version of the article.

Figure 1 does not have the standard caption format. The colors are too faint to distinguish among them. The legend, as well as city names on the map, are too small to be readable. Since most readers may not be familiar with the study region, including latitude/longitude coordinates would be a welcome addition.

Corrected. The modified text is in the new version of the article.

 Line 181 seems to have an extra indentation.

Ta uwaga nie jest zrozumiała, gdyż w wersji wysłanej formatowanie wydaje się poprawne.

The method section could be divided into "Data", "Statistical Analysis", etc. for increased readability.

Corrected. The modified text is in the new version of the article.

 Line 183 uses a symbol that does not look like "Y" in the equation.

Corrected. The modified text is in the new version of the article.

The results section is very difficult to follow. The first part of the results seems very disorganized with changing bolded/unbolded text.

Corrected. The modified text is in the new version of the article.

 While the Appendix is referred to in the Abstract (which should not be the case), it is not referred to before the tables in the result section. This is necessary because the variables are only defined in the Appendix and otherwise the reader cannot follow the results.

Corrected. The modified text is in the new version of the article.

In Table 1, the authors should consider listing the "Valid N" values together with a comma (no space in the thousands/hundreds) because it makes it difficult to read.

Corrected. The modified text is in the new version of the article.

 Line 281 has "sulphur" /"sulfur" misspelled.

Corrected. The modified text is in the new version of the article.

Line 295 has "obstructive" misspelled.

Corrected. The modified text is in the new version of the article.

 Line 343 and previous instances have a confusing grey font which makes it difficult to read.

Corrected. The modified text is in the new version of the article.

 I am not sure it is necessary to state that the source of data is "Author's own work" as that is implied. If it is not, that should be cited, but otherwise it is assumed the authors provided the data.

Corrected. The modified text is in the new version of the article.

 In Table 4, row 2, what is the cost unit for Pneumonia?

Corrected. The modified text is in the new version of the article.

 Lines 376 and 397 should be subsections for clarity. It seems that the authors have repeated information from the Discussion in the Conclusions.

Corrected. The modified text is in the new version of the article.

 For example future work should be better detailed and separated. 

Corrected. The modified text is in the new version of the article.

Submission Date

16 November 2019

Date of this review

28 Nov 2019 04:10:51

5.12.2019

5 Dec 2019

Round 2

Reviewer 2 Report

The authors have addressed the majority of the comments and suggestions. There are a few stylist issues that still need to be corrected.

Generally the authors have not subscripted numbers in pollutants such as the "3" in O3 or "2" in CO2 - this needs to be corrected for all pollutants. Table 3 could either have reduced sized font or be turned landscape because many of the variable names are cut off and need to go into another line. In many places, including Table 3, there seems to be discountinuity between using a "," or a "." as a decimal point (i.e. p column in Table 3). Table 1 could also be reformatted as either landscape or smaller font. The last two columns have stylistic issues - for example the distributions could simply be addressed as "log" or "log10" (with the 10 subscripted). Lines 282-284 show a bit of formatting carelessness in the part of the authors. These should be addressed throughout the manuscript.

Author Response

see the attahed file
